# 500-year paleoclimate record inferred from Greenland Juniper wood contextualizes current climate warming

Magdalena Opała-Owczarek [1] ✉, Ulf Büntgen[2,3,4], Piotr Owczarek [5] & Christian Lange[6]

Contextualising Arctic warming and sea ice loss requires high-resolution climate proxy archives, which are rare across the high-northern latitudes. Here, we present annually resolved and absolutely dated tree-ring width measurements of living and dry juniper (*Juniperus communis*) shrubs, as well as herbarium specimens, all from southern Greenland. We develop a continuous chronology for 1526–2023 CE that correlates at 0.67 with mean June–August temperatures ($p < 0.001$). We then identify reduced cell wall lignification (i.e., "Blue Rings") during exceptionally cold summers that often occurred after large volcanic eruptions, with Laki in 1783 and Tambora in 1815 CE causing the strongest responses. Our findings at the interface of dendroclimatology and wood anatomy provide a high-resolution paleoclimate record for southern Greenland that places anthropogenic warming in the context of natural climate variability.

The Arctic is warming at approximately 2–4 times the global average[1]. Known as Arctic Amplification, driven by human-induced greenhouse gas emissions, and manifested in the rapid loss of large ice masses[2], this trend is particularly strong during the past decades[3]. Understanding past temperature changes is essential to provide a baseline for assessing how the Arctic may respond to future warming. High-resolution climate proxy records are, however, sparse across the high-northern latitudes[4]. Greenland, with a large body of rapidly melting ice cap, plays a pivotal role in advancing our understanding of the impact of climate change on the natural environment, not only in the Arctic but also globally[5].

Climate reconstruction from Greenland primarily refers to using ice cores extracted from Greenland's ice sheets to understand past climate conditions[6]. These ice cores contain trapped air bubbles, isotopes, dust, and other compounds that can reveal information about environmental changes over time. Using ice cores has challenges, such as limited geographical coverage (ice sheet), dating uncertainties, and limited temporal resolution[7]. It should be, however, emphasised that

ice cores provide the most comprehensive record of volcanic eruptions that have contributed to periods of cooling[8–10]. The results from the sediment cores recovered from Greenlandic lakes provide an invaluable addition to the findings derived from the ice core samples. The continuous archive of past environmental changes based on these proxy data spans the entire Holocene[11,12]. Nevertheless, a potential limitation of these studies is the possibility of errors in the interpretation of [14]C and [210]Pb-dated sediment cores, involving e.g. problems of dating calibration and different sedimentation rates and disturbances[13,14]. Tree rings, one of the most widely used climate proxies, are invaluable and powerful sources of climate information due to their high resolution, accurate year-by-year dating, and the ability to reconstruct decadal to millennial-scale climate conditions[15]. However, the age of the tundra plants has limited the application of the dendrochronological method in Greenland. Existing shrub records of *Alnus viridis*, *Salix glauca* and *Juniperus communis* from West Greenland[16–20], *Salix arctica* from north-eastern Greenland[21] or from the only natural forest with *Betula pubescens* located in Qinngua Valley,

[1]Institute of Earth Sciences, Faculty of Natural Sciences, University of Silesia in Katowice, Sosnowiec, Poland. [2]Department of Geography, University of Cambridge, Cambridge CB2 3EN, UK. [3]Global Change Research Institute (CzechGlobe), Czech Academy of Sciences, 603 00, Brno, Czech Republic. [4]Department of Geography, Faculty of Science, Masaryk University, 613 00, Brno, Czech Republic. [5]Institute of Geography and Regional Development, University of Wroclaw, Wroclaw, Poland. [6]Natural History Museum, Copenhagen, Denmark. ✉e-mail: magdalena.opala@us.edu.pl

**Table 1 | Site locations and descriptive statistics of the juniper ring-width chronology from the Tunulliarfik Fjord area, southern Greenland**

| Sites | Narsarsuaq | Glacier Sermiat | Tasiusaq | Historical collection (Tasermiut, Igaliku, Qaqortoq) |
|---|---|---|---|---|
| Elevation | 80–100 m | 220–320 m | 150–200 m | – |
| Type of wood material | Living | Living/dry | Living/dry | Dry |
| Number of samples | 12 | 15/26 | 12/32 | 30 |
| Maximum shrub age | 200 | 368/267 | 314/264 | 354 |
| Chronology time span | 1526–2023 (498 yrs) | | | |
| Number of series included in chronology | 51 | | | |
| Mean length of series | 161.5 | | | |
| Average mean sensitivity | 0.462 | | | |
| Series correlation | 0.526 | | | |
| mean ring width (standard dev) | 0.400 mm (± 0.227 mm) | | | |
| Autocorrelation filtered/unfiltered | −0.029/0.553 | | | |
| Mean r-bar/EPS | 0.368/0.883 | | | |

southern Greenland[22,23] do not reach the pre-instrumental period. Until now, the existing dendrochronological data for Greenland have been insufficient for reconstructing the climate over the centuries.

Here, we present the combined dendrochronological and wood anatomical assessment of living and dead juniper wood from southern Greenland to reveal annually resolved and absolutely dated insights about natural summer temperature variability back into the Little Ice Age. This reconstruction extends beyond the instrumental period in Greenlandic terrestrial areas outside the ice sheet. Additionally, we aimed to assess the usefulness of the so-called "blue rings" as an indicator of post-volcanic cooling and as a key anatomical feature allowing for the synchronisation of juniper samples. Our study contributes to the understanding of the effects of volcanic eruptions on climate by combining wood anatomical observation with superposed epoch analysis to explore the significance of selected eruptions on growth ring width and blue ring formation. In order to more accurately evaluate the reliability of the reconstruction, we conducted thorough comparisons with other proxy data from southern Greenland and dendroclimatic reconstructions from the Arctic.

## Results and discussion
### Dendrochronological dating and construction of long-term composite juniper chronology
The oldest living shrub contained 367 growth rings, which, combined with historic and dry wood samples accounting for up to 354 years, made it possible to construct a reliable, well-replicated 498-year growth-ring chronology (1526–2023 CE). The mean width of annual growth was 0.400 (±0.227) mm/yr. Owing to the typical features of junipers, such as slow, irregular growth; eccentricity of growth; and high frequency of missing rings, juniper cross-dating was challenging; thus, from a total of 127 samples, only 51 samples were incorporated into the composite chronology (Table 1, Fig. 1). To obtain the best possible climate signal, only the well-correlated samples were selected. Statistical indicators of the quality of chronology, such as the average correlation between series (rbar) and expressed population signal (EPS), are variable over time but remain above the accepted critical thresholds (Fig. 2). Juniper is a very challenging species for dendrochronological investigations. The reliability of our dating is enhanced by implementing the serial sectioning method and several microscopic cross-sections, which significantly improve the legibility of extremely narrow growth, as well as the use of blue rings as time markers (Fig. 3). We detected 8 years with frost and 13 years with blue rings, most of which appeared after significant volcanic eruptions (Fig. 1). This was significantly less than in the few existing works on the occurrence of blue rings worldwide[24,25], making them unambiguous biochronological features. Difficulties in dating the juniper samples

and the rejection of half the samples were already indicated by Pellizzari et al.[26] and Carrer et al.[27] for alpine areas. In only a few dendrochronological studies to date, it has been possible to assemble a multi-century chronology of junipers from Arctic sites, of which the longest, based on both living and dry branches, comes from the Kola Peninsula (>500 yrs[28]), the Polar Urals (>600 yrs[29]), and northern Iceland (>800 yrs[30]). Recently, very old specimens of junipers have also been described for sites in northern Fennoscandia (Sør-Varanger, Kevo, Abisko) by Lehejček et al.[31] and Carrer et al.[32]. The assembled juniper chronologies for Greenland mainly concerned the area around Nuuk, where the relation to climate and Greenland Ice Sheet melting was elaborated over the last century[18,33]. Juniper discs with more than 200-300 rings were collected from southernmost Greenland during the scientific expeditions to Greenland at the end of the 19th century[34], and have recently been reconfirmed.

### Climate sensitivity
Although the usefulness of Greenland dendrochronological data as a climate proxy has previously been questioned[35], in our study, the significant climate sensitivity of juniper ring width was demonstrated (Fig. 4A, B). Unlike in alpine tundra, where juniper growth is mainly controlled by the amount of winter precipitation[26,27], the growth of juniper shrubs from the Greenlandic tundra zone is limited by the temperature in the current summer. Similarly, junipers from other circum-Arctic sites are temperature-stressed[18,29,30,36]; however, the strength of the signal varies depending on inter alia sampling strategy (adequate replication, proximity to the root collar) and site characteristics (homogeneity, geomorphic disturbances). To determine the suitability of juniper rings from southern Greenland as a climate proxy for reconstruction, dendroclimatic tests with several factors were carried out (Fig. 4A, B). The results primarily indicated time-stable, strong positive correlations between the growth ring index and summer air temperature ($r_{JJA}$ = 0.72, AD 1961–2023, $p < 0.001$). The positive correlation between winter-spring minimum temperature and juniper growth ($r_{Feb}$ = 0.38, $r_{Apr}$ = 0.34, 1961-2013, $p < 0.01$) corresponds with the negative correlation with snow cover, indicating that less snow cover and faster snowmelt have a positive effect on juniper growth. However, it should be noted, that precipitation was not statistically significant, as plants have access to moisture through both precipitation and high air humidity. So, long-lasting snow cover is unfavourable to juniper growth, thereby significantly shortening the length of the growing period during a short Arctic summer. These results concur with findings from other parts of Greenland where deciduous shrubs exhibit a consistent negative response towards the amount of snow precipitation[37,38]. A statistically significant dependence of the annual growth of junipers on cloud cover during the

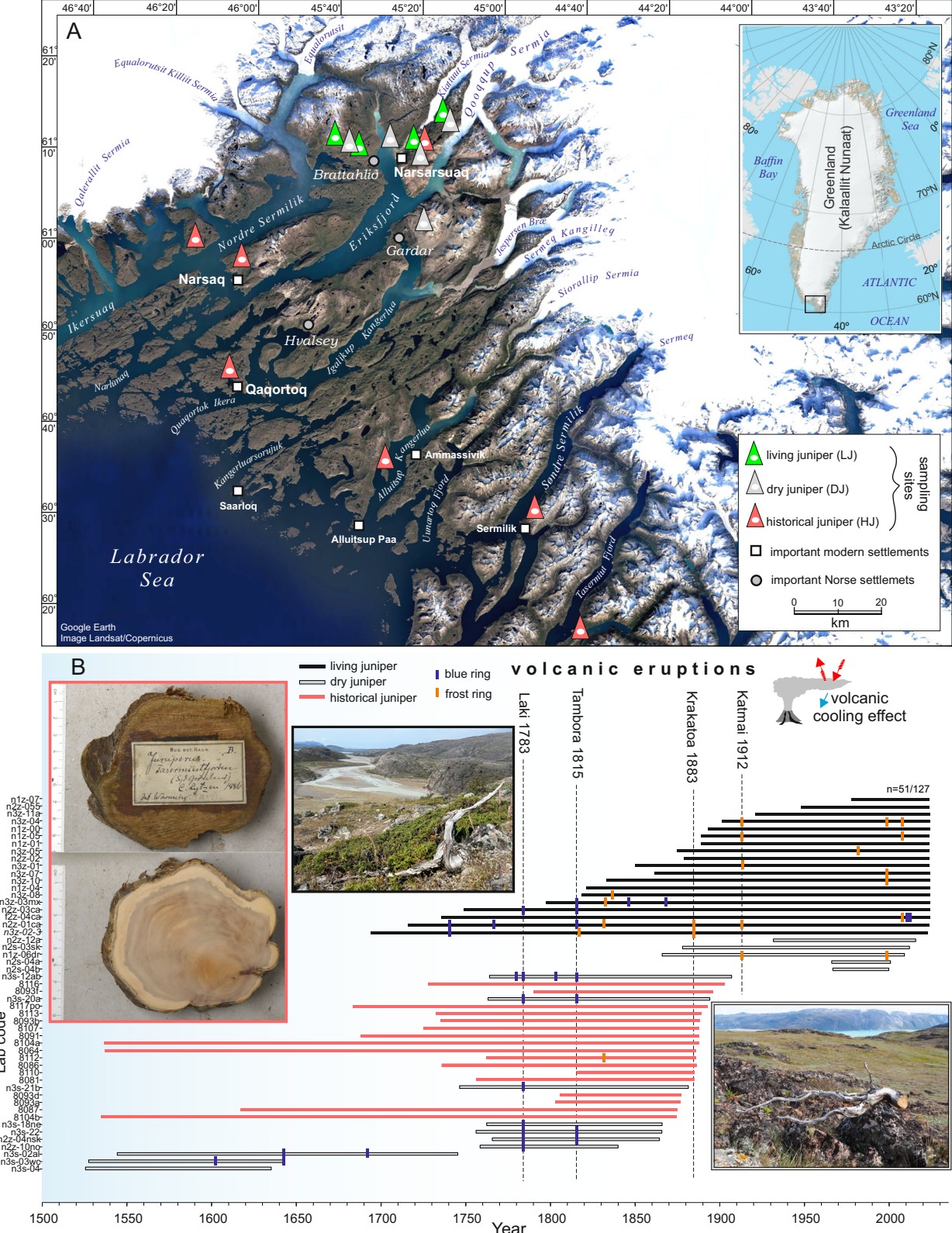

**Fig. 1 | Southern Greenland sampling sites and juniper growth-ring data.**
**A** Location of the study area in the southernmost part of Greenland with sampling sites, **B** Dating of juniper dendrochronological sequences showing the time span of each sample, with the occurrence of blue and frost rings resulting from the main volcanic eruptions marked, and examples of living, historic and dry wood material.

From the total of 127 samples, 51 were used for climate reconstruction. The base maps were downloaded from Google Earth and GEUS (Geological Survey of Denmark and Greenland, Topographic Map of Greenland); the images of historic juniper cross-sections: © CC BY-NC 4.0 Natural History Museum of Denmark, University of Copenhagen.

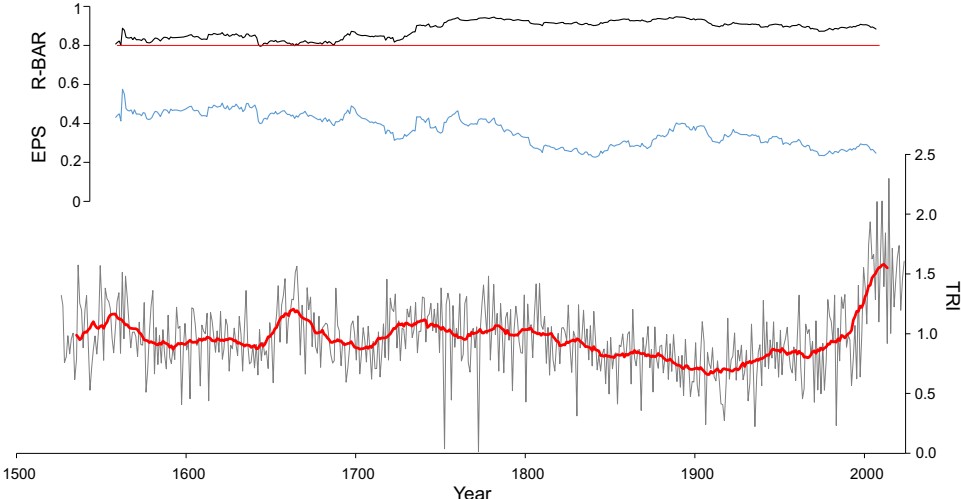

**Fig. 2 | Annually resolved growth-ring chronology from the southern Greenland over 498 years.** Common juniper standard growth-ring-width chronology (grey line) with smoothed values (21-year running average - red line) covering the years 1527-2023, and it expressed population signal (EPS) and correlation coefficient (Rbar) statistics. The EPS value exceeded the threshold of 0.80 over the full time span of the chronology, as indicated by the red horizontal line.

winter months was found ($r = 0.32$, 1961–2013, $p < 0.05$), as the strong radiation of heat from the ground during cloudless periods, especially in winter, led to large temperature decreases. This was also confirmed by the results of the negative effect of low winter temperatures on juniper growth.

Sea ice loss may have a dominant effect on local coastal temperatures but a small contribution to overall warming in Greenland[39]. Given the lack of complete consensus on the effects of sea ice loss on shrub growth, we investigated the relationships between juniper growth and sea ice extent (Fig. 4C). The results of the present study revealed an increasing trend in juniper growth with decreasing sea ice extent ($r = -0.59$, 1901–2015, $p < 0.001$), particularly during notable recurrent periods of significant ice cover increases in 1907, 1935–1936, 1983–1984 and 2008, resulting in low growth. However, attempts to reconstruct this element have been unsuccessful and require future study, as sea ice extent does not necessarily have a simple linear relationship with summer temperature.

**Five-century-long reconstruction of summer temperatures**

A mean summer temperature reconstruction from 1526 to 2023 CE was obtained via the linear regression equation. The coefficient of determination was 0.52, indicating that 52% of the variance was predicted by the model, which passed commonly used calibration and verification statistics (Fig. 4D-F). The reconstructed mean annual temperature during the last 498 years was 9.4 °C, similar to that of the reference period of 1961–1990 (9.3 °C), but much warmer in the 21st century (10.6 °C) (Fig. 5). The long-term trend in the five hundred-year-long June–July–August (JJA) temperature reconstruction was insignificant. However, there has been a warming trend over the last century, especially since the second half of the 20th century and throughout the first decade of 21st century (the mean annual temperature trend +0.15 °C decade⁻¹, the JJA trend was 0.25 °C decade⁻¹ ($p \leq 0.01$) from 1961–2010). The meteorological and tree ring data are consistent and show that the southern part of Greenland experienced decadal periods of both cooling and warming during 1961–2023, with an inflection point around the mid-1990s, and no significant warming after ~2010 (Fig. 6). The reasons for this variability can be found in the regional influence of the large-scale circulation, represented by the indices for the NAO (North Atlantic Oscillation) and GBI (Greenland blocking index). Zhang et al.[40] concluded that, since 2011, there has been a shift towards a more negative phase for GBI and a more positive phase for NAO, but at the

same time, these have been highly variable, with more modest values and reduced warming recently.

Similarly, a warming trend was recorded only during the decade of 1650–1660, but the summer temperatures in the 2000s and 2010s were significantly higher than those in previous centuries (anomaly of +1.5 °C). These results had significant environmental consequences, as each 1 °C increase in summer temperature corresponds to approximately 116 Gt*yr⁻¹ of GrIS (surface) mass loss and a 26 Gt*yr⁻¹ increase in solid ice discharge[41]. Recently, using ice core-based temperature reconstruction, Hörhold et al.[42] reported that the warming in central and northern Greenland in the first decade of the 21st century exceeded the range of preindustrial temperature variability in the past millennium. Even though warming has not progressed at the same rate across the island, on the basis of dendroclimatological reconstruction, it can be concluded that the warming in southern Greenland in the first decades of the 21st century has been unprecedented over the last 500 years. Strong warming in the 20th century, which contrasts sharply with the preceding cooling trend, was documented in the Arctic on the basis of multi-proxy records; however, records from Greenland were mostly from glacier ice[43].

It was assumed that the Little Ice Age (LIA) for the Northern Hemisphere occurred over a period of approximately 480 years, spanning AD 1440–1920. The available proxy data suggest that the LIA in Greenland began in approximately 1250 AD, peaked in the 15th century, and ended in the mid-18th to early mid-19th centuries. However, the exact timing of individual fluctuations remains problematic due to poor chronological resolution[44] or asynchronous advances of mountain glaciers during the LIA. Interestingly, the glaciological evidence for southernmost Greenland, i.e., the ¹⁰Be surface exposure age of the prominent Narsarsuaq moraine, does not confirm that the LIA was the most notable period, thereby dating the maximum extent of the Kiagtût Sermiat glacier to much earlier, i.e., to the late Holocene pre-LIA maximum[45].

Owing to juniper growth-ring-based JJA temperature reconstruction, we accurately traced summer temperature fluctuations in the southern fjords of Greenland from the present to the mid-16th century. While we could not pinpoint the start of the LIA, three periods were identified: the first was a long cool period (1570–1650) preceded by brief warming; the second was a period of highly dynamic cool and warm periods (1651–1815); and the third was another long cool period ending the LIA at the beginning of the 20th century (1816–1918), when the long-term trend of rising temperatures began. In light of the

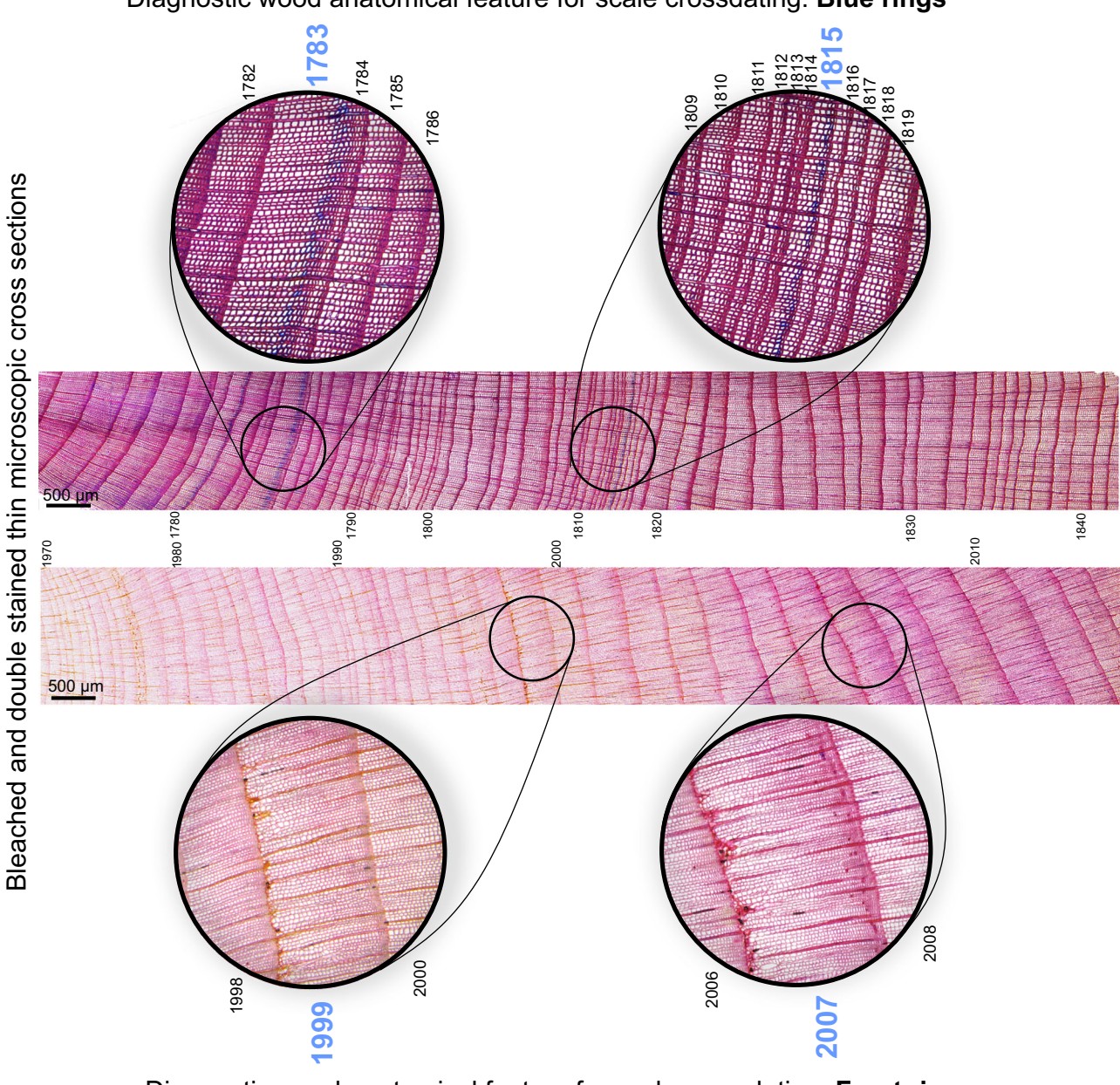

**Fig. 3 | Wood anatomical anomalies in juniper from southern Greenland.**
Examples of anatomical thin sections showing the blue and frost rings, as markers for samples synchronization. In total, we detected 13 years with BR and 8 years with FR, of which BRs were most common in: 1815 (58.3%, $n = 12$) and 1783 (72.7%, $n = 11$), and FRs were most common in: 2007 (13.6%, $n = 22$), 1999 (16.6%, $n = 24$), 1912 (27.7%, $n = 18$); where n indicate available number of samples in the analysed year. It should be noted that the occurrence of BRs and FRs does not apply to herbarium samples from which thin sections were not made, nor to low-correlated samples, which were not used in climate reconstruction.

dendrochronological data, it should be concluded that the LIA was not a period of uniform cold weather in southern Greenland. We found substantial increases in temperature in approximately 1550–1570, 1650–1670, and 1740–1750, but the warming of the early 20th century did not appear very clearly. Particularly interesting was the reconstruction of the solar activity-related cooling during the Maunder Minimum (MM) period (1637–1717), which was a very dynamic period in southern Greenland. The early MM started as a very cold decade and then was split by the warmest two decades, 1654–1674, and the late MM showed evident strong cooling, culminating in the early 18th century. The following century was a period of climatic fluctuations with distinct cooling after the eruption of Tambora in 1815. Another very cool period was the turn of the 19th and 20th centuries, with the lowest temperatures occurring from 1915 to 1917.

A comparison of the dendroclimatic reconstruction from unglaciated coastal areas of southern Greenland with other climate proxy-based reconstructions (marine sediment core, ice core, and lake sediment) and historical climate data[46] revealed similarities in the climate history of the southern coast of Greenland (Fig. 7). In the closest vicinity of the juniper study sites, climate variability was previously inferred from Lake Igaliku sediments[47]. A comparison of the data obtained from the chironomid assemblages with the juniper-based reconstruction reveals a number of similarities, except for the periods between 1640 and 1790, and between 1920 and 1970. The most

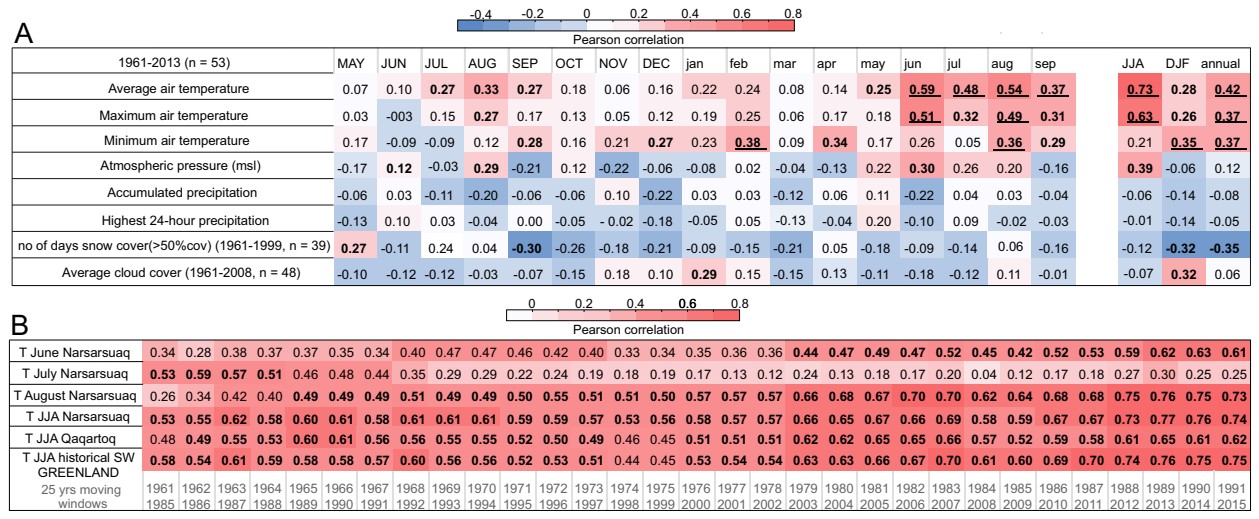

**Fig. 4 | Assessment of the climate sensitivity of the juniper dendrochronological record. A** Correlation coefficients between *Juniperus communis* growth and Narsarsuaq climate data from May of the previous growing season to September of the current growing season. Colours indicate Pearson's correlation coefficients calculated between chronology and selected climate variables, with statistically significant r values marked. Levels of significance indicated by bold values for $p < 0.05$ and underlined values for $p < 0.01$. The window of analysis varies depending on the availability of the climate record. Correlation coefficient and significance value for the June-August temperature are $r = 0.726$ and $p = 1.6e-11$ (two-sided t test, 1961–2023). **B** Moving correlation functions between growth-ring chronology and summer month temperature from different climate records from

S Greenland with 25-year window, significant correlations ($p < 0.01$) are indicated by bold values, **C** Comparison of the annual sea ice extend records with growth-ring width variability showing clear relationships during periods of extremes (grey shadings), and the dotted line represents the interannual trends, **D** Split sample calibration/verification of the reconstruction model based on early (1961–1992) and late subperiods (1992–2023). All evaluation statistics reach a significance level of $p < 0.01$, indicating statistical skill in the regression model, **E** Full period calibration of the model (1961–2023) complemented by measures of reconstruction skill (r, RE, CE, and ST–see the text for explanation), **F** Scatter plot illustrating the regression used to reconstruct summer temperature.

consistent signals are the strong modern warming and the relatively warm conditions during the Dalton Minimum and the warming of the early 17th century (Fig. 7E). There was also a remarkable correspondence between the diatom-inferred August sea surface temperature from the marine sediment core[48] and the terrestrial record of *Juniperus communis* growth rings (Fig. 7B). Common cooling occurred at the end of the 16th century, followed by abrupt, significant warming in the mid-17th century, a colder turn in the 17th and 18th centuries, and a relatively warm turn in the 18th and 19th centuries, culminating in marked cooling concentrated in the mid-19th century. Most of these fluctuations, especially the pronounced warming of the mid-17th century, were also exhibited by ice core-based reconstruction[49] (Fig. 7D). The least concordant was the compilation of the 59 proxy data for the Arctic[50], although the high degree of concordance in the 16th and 17th centuries was noteworthy (Fig. 7G). Interestingly, this strong mid-17th-century warming in all the proxy data from southern Greenland was quite pronounced, and the warming is also known from early instrumental data from England and Western Europe.

Comparisons of the new dendrochronological record from southern Greenland with dendrochronological data from the northern tree lines of Eurasia and North America revealed similarities and significant differences in pan-Arctic temperature variability (Fig. 8). The greatest agreement was found in the Northern Quebec temperature reconstruction[51], which shows similar fluctuations in both warm (recent warming; turn of the 18th and 19th centuries; first half of the 18th century) and cool (1840s–1950s) periods. By contrast, the common warm periods in Greenland and Alaska[52] are the recent warming period, the 18th century, and the mid-16th century. The most notable inconsistency concerns the Maunder Minimum period, which is distinctly cold in many reconstructions, but not clearly marked in Greenland. Similarly, this MM cooling is absent in data from the Russian Arctic[29]. The reconstruction of temperature variability from the Kola Peninsula[53] differs most significantly, with only the warming and subsequent cooling in the 17th century being consistent with that observed in southern Greenland (Fig. 8).

## Extreme years

Extreme years are manifested by the occurrence of narrow rings or disrupted cell formation, such as BRs and FRs (Fig. 3). Most of these extreme years, associated with volcanic eruptions, are consistent with studies carried out at the northern tree limit in both Asia and North America, e.g. 1600, 1641, 1783, 1815-1817, 1868, 1884, 1912, 1916-1917[54–58]. Interestingly, the cooling after the Laki and Tambora eruptions was visible in the reconstruction as a pronounced temperature drop 1 year after the eruption, but not the largest occurring at the 500-year scale analysed. Similarly, the Katamai eruption barely registered at all. However, the record of these events in wood anatomy was very clear (FR in 1912; BR in 1783 and 1815). In total, we detected 8 years with FR and 13 years with BR, of which BRs were most common in 1783 (33%) and 1815–16 (24%) (Fig. 5A). The growth-ring width integrates favourable conditions of a longer growing season, whereas BR and FR wood anomalies reflect rapid and short-term temperature drops. For example, the FRs in 1999 and 2007 were associated with daily temperatures below freezing in May. However, determining the meteorological reasons for the formation of blue rings is difficult, as they occur mainly during periods for which no instrumental data are available. Interestingly, BRs were more common in the earlier period (and dry wood samples) than in the modern period, when FR mostly occurred. In general, BRs appear to occur during warm periods, such as the 1720–1820 period and the modern warm period. In contrast, FRs occur more frequently during the cool episodes of the Little Ice Age. These results corroborate the hypothesis that prolonged cooling does not diminish lignification but rather increases the likelihood of frost-related damage. These observations coincide with the results of studies conducted at the upper tree line in the Swiss Alps[59].

We evaluated the accuracy of our dendrochronological data by comparison with volcanic signals from the Greenland ice sheet[10]. Volcanic aerosol peaks contained in the ice core (NEEM-2011-S1) are consistent with dendrochronological signals (narrow rings, BRs, FRs), which are closely related to major volcanic eruptions (Fig. 5 A, B). This is particularly pronounced in years of volcanic eruptions: Huaynaputina (1600), Mt. Parker (1641), Hekla (1963), Laki (1783), Tambora (1815), Mt.Katmai (1912). These events were followed by the occurrence of BRs, FRs or strong growth reduction in tree-ring records. Superposed epoch analyses show a clear post-volcanic cooling signal after most major eruptions, or with a response lagged by one year (Fig. 5C). Of particular interest is the response of Greenland's junipers to the major eruptions of Laki and Tambora, where BRs were recorded in the year of the eruption and narrow growth a year later (Fig. 5Ca).

## Imprint of the Laki (1783–1784) eruption on the dendrochronological record from Greenland

The 8-month Laki eruption in southern Iceland, from 8 June 1783 to 7 February 1784, was one of the largest and best-documented eruptions in historic times. It produced the second-largest basaltic lava flow and the second-largest pyroclastic fall deposit from an Icelandic eruption. The Laki eruption had a major impact on Northern Hemisphere temperatures, causing a cooling of 1–2 °C below average. The year 1783 was referred to in Europe as 'Annus Mirabilis', and there are many historical records (weather logs, diaries, scientific publications and newspaper articles) of the occurrence of phenomena such as widespread sulfuric aerosol clouds, referred to as dry fog and haze in chronicles in Europe at this time[60].

The initial appearance of the Laki haze is well documented. The eruption was most intense during the first 1.5 months. Analysis of synoptic weather maps suggests that bulk transport occurred at the level of the polar jet stream, and those that reached Greenland were most likely tropospheric; they were first transported east or northeast by southwesterly winds and then sank and deposited material on the Greenland ice cap after circumnavigating the polar region[60]. Several observations from Iceland, Greenland and other Nordic countries were recorded at the end of June 1783. Smoky haze was observed around Nuuk (Godthab) in Greenland, where, according to the Ephemerides Societatis Meteorologicae[61], "the sky was covered with clouds resembling smoke". This coincides with the immediate appearance of the Laki signal in the early summer snow of 1783. Highly elevated acidity levels were detected in the Greenlandic ice cores[62]. Unfortunately, the available meteorological data (SW Greenland series) do not cover 1783 or the first half of 1784. However, the average monthly temperatures for the second halves of 1784 and 1785 were significantly lower than the average for the period of 1900–2020.

A dendrochronological and cellular-scale record from Greenlandic Juniper facilitates the cell-by-cell tracing of this extraordinary Laki event (Fig. 3). The annual growth in 1783 was relatively wide (comprising approximately 10–15 rows of cells), and the growth of shrubs started as it would have in any "normal year". However, following the eruption of the Laki volcano, the final rows of cells exhibited incomplete lignification of the cell walls, as evidenced by the presence of the blue ring in the dyed wood samples. This phenomenon serves as an exemplary illustration of the efficacy of the BR as a sensitive indicator of a sudden temperature decrease following a major eruption. Intriguingly, the subsequent year of 1784 was characterised by a remarkably narrow ring, which was a consequence of the deterioration of weather conditions that ensued after the eruption. This record stands as the sole explicit dendrochronological documentation of extreme environmental conditions following the Laki eruption, with prior research indicating that ring width values for 1783 are not atypical for treeline sites in Alaska and the Yukon Territory. However, analysis of wood density has attributed the unusually low density and lighter appearance to the cold, volcanic summer of 1783[63]. Newer

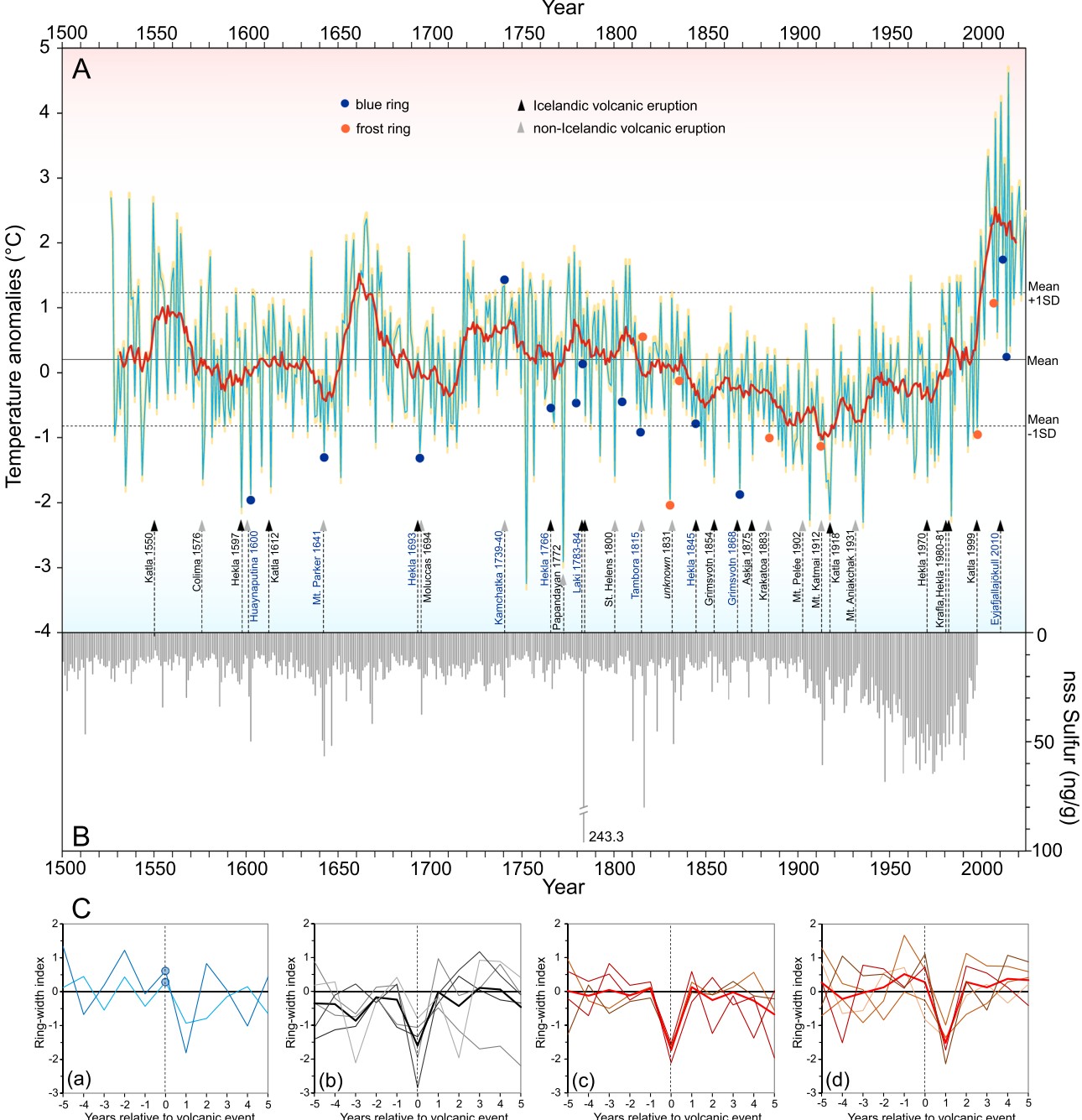

**Fig. 5 | Summer temperature reconstruction based on juniper growth-ring from southern Greenland and volcanic signal comparison.** A 498-years-long reconstruction of summer temperature variations (June-August) (as temperature anomalies from the reference period 1961–1990). The overall mean of the reconstruction is depicted by the horizontal black solid line, and black dashed lines indicate the bounds of one standard deviation from the mean. Notable blue and frost rings marked against the background of major Icelandic and global volcanic eruptions compared with (**B**) ice-core inferred non-sea-salt sulphur (nssS) records from Greenland (NEEM-2011[10]) and (**C**) growth-ring width response to volcanism since AD 1550: a superposed epoch analysis (SEA) of (a) Laki and Tambora eruptions having the most pronounced BR signal in the year 0, (b) global volcanic events (VEI > 4) (1600, 1772, 1809, 1883, 1902, 1931), (c) Icealndic volcanic eruptions with the signal in the year 0 (1597, 1854, 1868, 1970), d) and in the year +1 (1550, 1612, 1693, 1766, 1981).

research, on the basis of anatomical analysis of wood from Alaskan trees, confirmed a very thin cell wall in the ring of 1783[64]. It is hypothesised that the Inuit 'Time Summer Time Did Not Come', described in oral history as a disaster caused by extreme cold and an abrupt end of the warm season after June passed, with no warm weather until the next spring, occurred in 1783 because of the eruption of Laki[63]. According to Jones et al.[65], the years 1783 and 1784 are not considered extremely low density for Europe. The limited work on the occurrence of blue rings shows no wood anatomical effects of the Laki eruption in

1783–1784, a small signal following the Tambora eruption in 1815, a clear record of the Samalas eruption, and a highly visible cluster of BRs near 536 CE[24,25,66].

In conclusion, the results extend the research potential of dendrochronology beyond the northern tree line into polar tundra areas, which have not been entirely represented in long-term dendroclimatological reconstructions. The developed record is one of the longest growth ring chronologies for Arctic shrubs to date. We reconstructed a half millennium of summer temperatures in southern

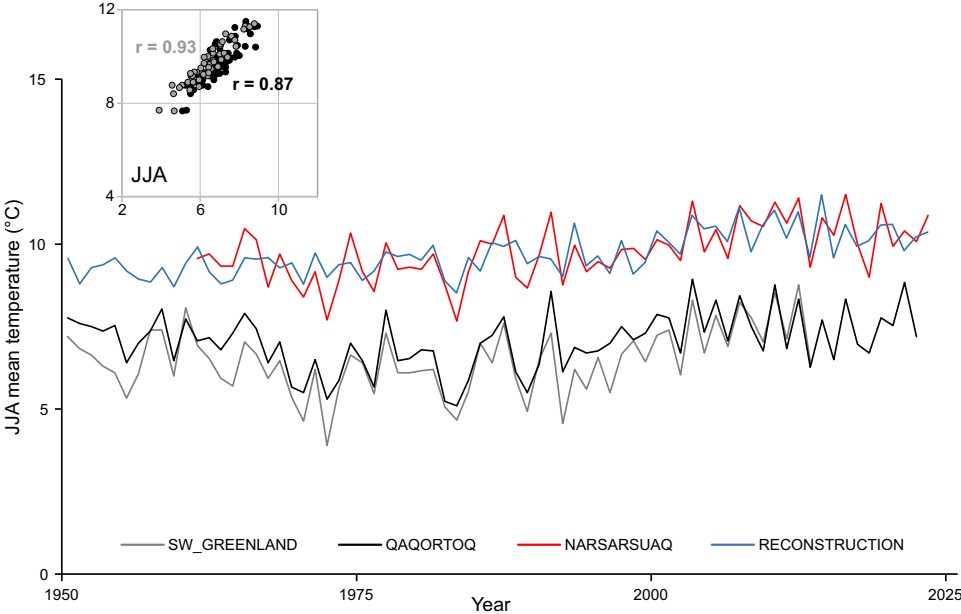

**Fig. 6 | Comparison between reconstructed and instrumental summer temperature time series over the 1950-2023 period.** Summer instrumental temperature data from Narsarsuaq, Qaqortoq and compilation of different series from SW Greenland compared with dendrochronological reconstruction of JJA temperature. The scattered plot shows the degree of similarity between the Narsarsuaq and Qaqortoq temperature time series (black dots) and the Narsarsuaq and SW Greenland temperature time series (grey dots).

Greenland using annual ring chronologies and wood anatomical anomalies of living, dry and historical junipers. The unique results obtained through the use of specimens from historical scientific expeditions, currently preserved in herbaria, highlight the importance of reanalysing old botanical collections using modern analytical techniques. Since the area of our research, located within Eriksfjord, was inhabited by the Vikings from 985 CE to the mid-15th century (Eastern Settlement), research into past climate change is becoming increasingly important. Integrating wooden archaeological finds from this area with the established five-century-long juniper growth ring chronology will be the next step in reconstructing past changes in this area over a period of environmental changes at the turn of the 15th and 16th centuries.

We have made the absolute dating of extremely challenging research material more reliable by using not only a classical cross-dating method but also independent chronological markers—blue rings—a thermally induced, disturbed cell wall lignification visible on double-stained thin sections of wood. The analysis of these anatomical changes has allowed for a significant deepening of conclusions. Wood samples from southern Greenland made it possible to study the effect of a large explosive eruption on the anatomical structures of wood. Despite numerous tephra deposits, meteorological data and historical accounts indicating a harsh summer in 1783 and a severe winter 1783/1784, the blue rings are the first biological evidence of climate disturbances associated with the Laki volcano eruption in the Arctic. Detecting these anatomical features will provide a more complete and accurate understanding of the eruption's dynamics and atmospheric effects, and will help to create a useful model for assessing the impact of past volcanic activity on the Earth's climate.

This study has important implications for palaeoclimatological research in the Arctic and adjacent regions in the context of ongoing climate change. By providing a 500-year context for long-term climate variability, it has been confirmed that the observed recent warming is unprecedented, even in the southernmost part of Greenland. The remarkable consistency of the magnitude of the mid-17th century warming and the timing of cold and warm subperiods during the LIA was found with other biological proxy-based data from southern Greenland.

## Methods
### Permits and permissions
The historical juniper specimens originate from the Herbarium at the Natural History Museum, University of Copenhagen, Denmark. Permission to conduct the museum query was granted by Collection Manager Olof Ryding and Collection Assistant Karen Bach in 2021. No destructive sampling was performed. The research used high-resolution photographs of juniper cross-sections collected during historical polar expeditions, which were made available by the collection manager of the Greenland Herbarium at the Natural History Museum of Denmark in Copenhagen. In the field, samples were collected outside protected areas and archaeological sites.

### Site description
The sites selected for this study are located within the extensive ice-free land area (c. 60–61°30'N, 44°30–47°W) near the southernmost part of the Greenland ice cap, rendering it a pivotal location for climate variability research. The ice sheet margin is approximately 100 km inland to the northeast of the Labrador Sea coastline, whereas the ice-free zone extends between 150 and 200 km in width between the southwest and southern lobes of the ice sheet. The area is distinguished by its high landscape diversity. In the eastern and northeastern parts of the area, there are high mountains up to 2000 m above sea level with numerous cirque and valley glaciers. Lowlands are found mainly in valley zones, with the broadest valleys around Narsarsuaq and Igaliku. The varied landscape results in very high spatial variability in climate conditions and vegetation cover. The mean annual air temperature in Narsarsuaq is 1.1 °C. The mean summer temperatures markedly differ between the outer coast and inland areas. The interior of southern Greenland has a subarctic, subcontinental climate with the highest temperature of 10.8 °C in July (Narsarsuaq) and a yearly precipitation of approximately 600 mm. Strong, desiccating warm and dry Foehn winds from the ice cap are frequent all year round. In winter, this wind causes sudden warming and melting. The temperatures are lower near the coast because of the cold sea. The average July temperature in Qaqortoq is 7.6 °C, with a yearly precipitation of 900 mm.

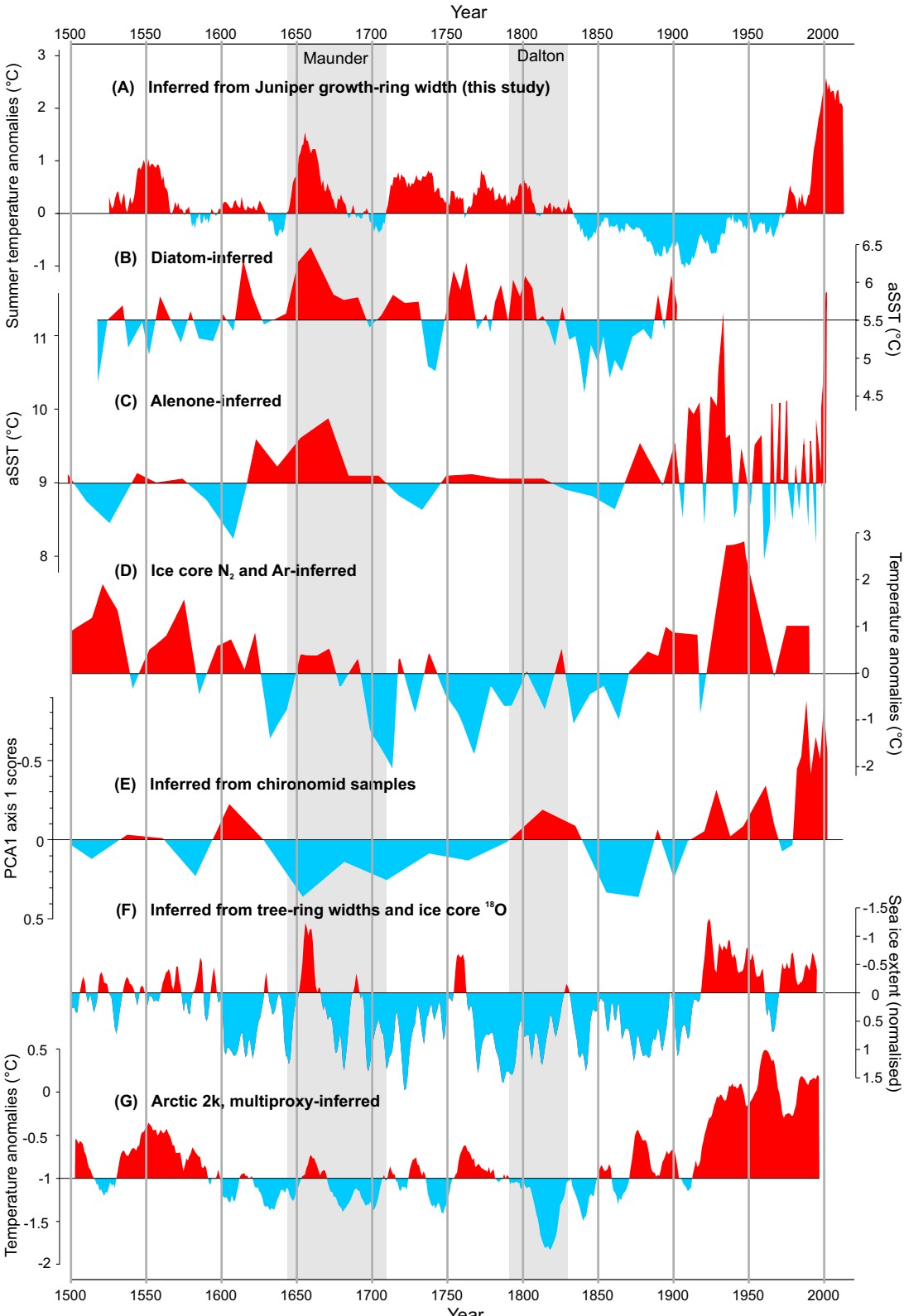

**Fig. 7 | Comparison of paleoclimate terrestrial and marine records from south and southeastern Greenland. A** Juniper growth-ring-based summer temperature reconstruction (this study), **B** Diatom-based August SST reconstruction from SE Greenland (Kangerdlugssuaq Trough)[49], **C** Alkenone-based SST reconstruction from a sediment core from SE Greenland (Sermilik Fjord)[83], **D** Surface temperature reconstruction derived from isotopes of $N_2$ and Ar in air bubbles in an ice core[50], **E** chironomid stratigraphy (PCA axis scores) in a sediment core retrieved from Lake Igaliku[48], **F** Greenland Sea winter sea-ice extent reconstruction of the Western Nordic Seas based on tree-ring widths and ice core ¹⁸O records from Fennoscandia and Svalbard[85], **G** Arctic temperature reconstruction based on 59 multiproxy records from north of 60°N[51].

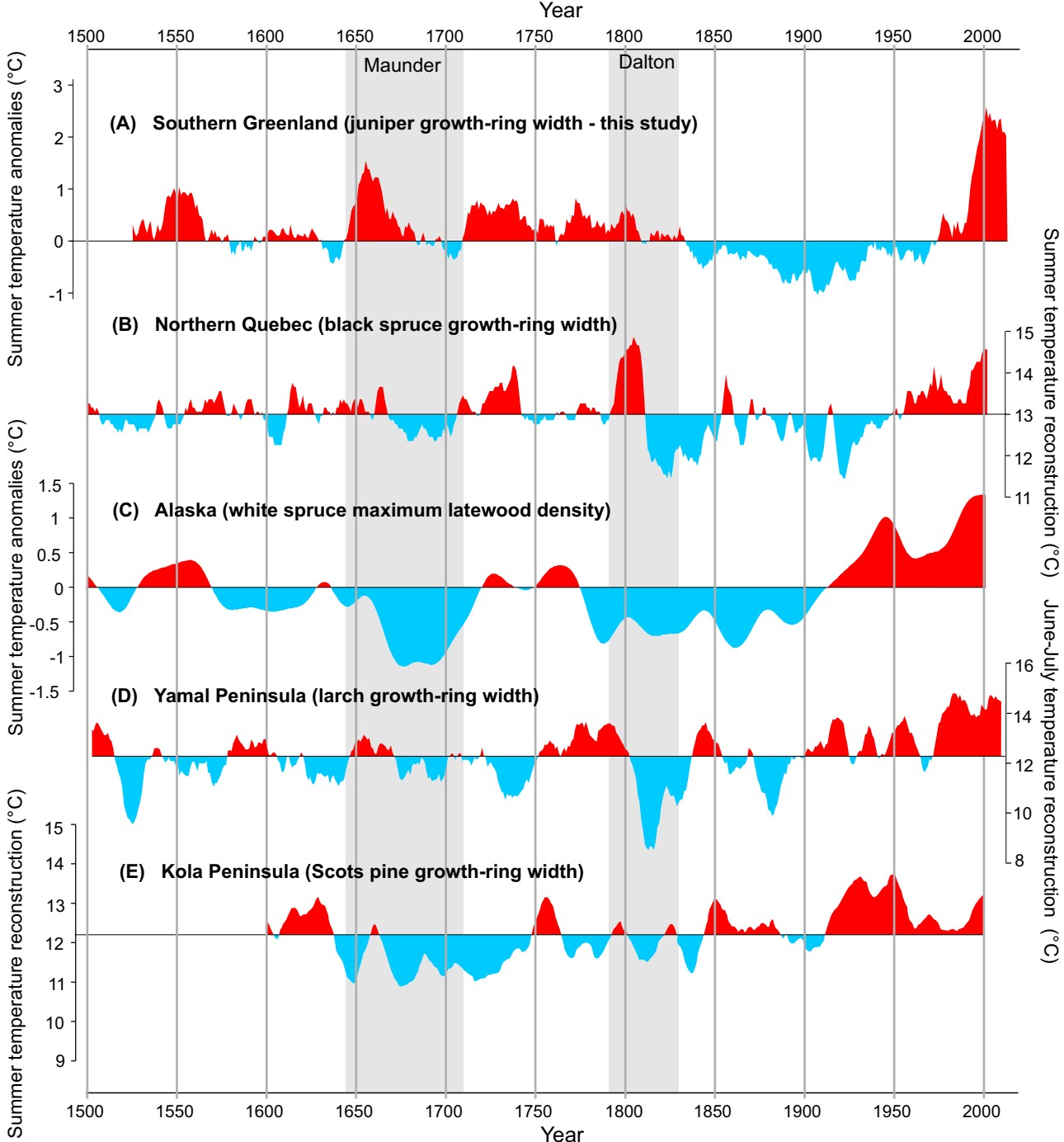

**Fig. 8 | Comparison of summer temperature dendroclimatic reconstructions from different parts of the Arctic. A** Juniper growth-ring-based summer temperature reconstruction (this study), **B** black spruce growth-ring-based summer temperature reconstruction from northern Quebec[52], **C** summer temperature reconstruction from a white spruce maximum latewood density from northern Alaska[53], **D** reconstruction of June–July temperature based on Siberian larch data from Yamal Peninsula[85], **E** reconstructed July–August mean temperature variations based on the Scots pine ring widths from Khibiny Mountains, Kola Peninsula[54].

Favourable climatic conditions influence the occurrence of rich tundra and subarctic communities. Dwarf-shrub heath is the most widespread plant community and is dominated by *Betula glandulosa*, *Vaccinium uliginous*, *Empetrum nigrum* and *Salix glauca*. *Juniperus communis* can be found both in coastal areas and inland fjords and plateaus. Willows cover relatively large areas in lowlands, especially on slopes with abundant snow cover during the winter. Ground vegetation is commonly dominated by different species of ferns, grasses, or forbs, such as *Ranunculus acer* and *Lathyrus japonicus*[67]. *Betula pubescens* constitutes open woodlands or

thickets in the most favoured places, among others, in sheltered parts of the valley near the Narsarsuaq.

## Wood material

*Juniperus communis* L. subsp. *nana* Syme. of the *Cupressaceae*, the only indigenous conifer in Greenland, occurs sporadically in the SE and SW parts of the island. It has prostrate growth with spreading branches but occurs most frequently as espalier forms, growing preferably over rock surfaces and against upright rocks. This low shrub, which is up to 50 cm tall and has a stem diameter rarely exceeding 10 cm, has a relatively long

lifespan. Although early work on juniper from southernmost Greenland indicated its potential longevity, very limited tree-ring research has been undertaken in the study area. C. Lytzen indicated that the oldest juniper samples may be as old as 380 years, and this is the first published scientific information on the age of juniper from Greenland[34]. J. Lange[68] and C. Kruuse[69], who worked comprehensively with juniper samples from this area, presented the characteristics of the species, highlighting the varying degrees of eccentricity and the small amount of wood produced each year, but of comparatively old age. Similar observations were subsequently performed on juniper wood collected by Daniels, De Molenaar, and Ferwerda during the Dutch Botanical East Greenland Expedition in 1966. Importantly, the phenomenon of cambium dormancy, cessation of growth in the upper part of the branch, and sometimes partially dying off, was pointed out, while the area deeper under the snow continues to grow. Such interruptions lead to the production of discontinuous growth rings[69,70]. Additionally, studies by Miller[70] demonstrated that the stems of juniper from West Greenland were indicators of its age of at least 200 years, which was also confirmed later by Buras et al.[33] and Lehejcek et al.[18], who examined the potential of shrub ring width and wood anatomy as proxies for climate and GrIS-melt. Tumajer et al.[71] point to the intensification of growth rates during the short growing seasons in the Arctic.

### Historical and field dendrochronological data

In August 2021, through the search for and exploration of the resources of the Greenland Herbarium in Copenhagen, housed at the Natural History Museum of Denmark, fifty-six museum artefacts and pieces of juniper wood were found that were collected during past scientific expeditions to Greenland. Most of the historical botanical samples had brief metadata on the collector, date and location of sampling, which is crucial for further dendrochronological analyses[72]. The samples came from 13 locations, including southern Greenland (Tasermiut, Igaliku, Qaqortoq, Sermilik, Unartoq), and were collected from 1873 to 1902 by C. Kruuse, E. Warming, C. Lytzen, N. Hartz and anonymous collectors (Supplementary Table 1). A detailed report on these research expeditions contains an interesting description of the juniper samples that were collected; this description includes the number of rings and their average width[34].

The labels on the herbarium specimens were used to identify the exact locations where they were collected at the turn of the 19th and 20th centuries, so that contemporary samples could be taken from the same area. Living and dry juniper samples were collected in southern Greenland in the upper parts of the Tunulliarfik Fjord (Eriksfjord) during the field research campaign in the summer of 2023. The sampling sites were located at elevations of 180–280 m a.s.l. in two key areas: the northern part of the Akuliaruseq Plateau, northeast of Narsarsuaq, between the Kiattuut and Qooqqup outflow glaciers, and the hilly area of Sammisup Timaa to the north of Qassiarsuk (Supplementary Table 2). In general, sampling was random, taking into account the location of the plant to ensure site homogeneity and to exclude the influence of abiotic factors on growth, such as mass movements. Particularly noteworthy was the discovery of a large amount of dead, dry wood collected for dendrochronological studies. In most cases, these dry samples were found to be attached to the ground with remnants of the root system, indicating that they were discovered in situ. Straight sections of the main prostrate stems were sampled for analysis to exclude the possible presence of reaction wood. Samples were also taken from contemporary growing specimens for calibration with meteorological data. For these specimens, discs with a thickness of 2-3 cm were collected from one of the main stems to avoid destroying the entire plant.

### Laboratory analysis and construction of the chronology

Two analytical paths were adopted: one for the historical samples and another for samples collected in the field. A total of 127 samples were analysed, 92 from the field and 35 from the herbarium. Owing to the great value of historical samples, measurements were made on the basis of high-resolution scans. For these samples, it was impossible to carry out invasive examinations, i.e., serial sectioning or microscopic cross-section preparations. The scans were taken from the original sanded surface prepared by the collectors and botanists, who analysed the number of rings on these samples at the turn of the 19th and 20th centuries, and the images were subjected to further analytical processing. This noninvasive approach has limitations; therefore, only 35 usable images were obtained from all 56 historical samples.

A comprehensive anatomical investigation was conducted on both living and dry juniper samples collected in the field. For each type of juniper sample, depending on the length of the shoot collected in the field, 2–3 anatomical thin sections that were 15 μm thick were prepared[73] using a rotary microtome (Leica RM2125). At least two repetitions were performed for each cross-section. Anatomical wood features were emphasised using a complex laboratory procedure. In the first step, each sample was bleached with sodium/potassium hypochlorite to decolorize the cell walls and to remove undesired cell contents[24]. Next, we applied the double-staining procedure to all microsections using Safranin (red) and Astra Blue (blue) dyes to visualise lignified and less lignified cell walls in red and blue, respectively[73]. The samples were then washed with distilled water, dehydrated with increasing concentrations of ethanol, and permanently fixed via an Euparal mounting medium on single microscope slides. Permanent thin sections were dried for 12–24 h at 60 °C. All microslides were photographed using a Leica DFC420C camera connected to a Leica DM4500B microscope. Leica LAS-X was used for image acquisition. Multiple images taken at 10× magnification were subsequently stitched together using PTGui software (http://www.ptgui.com, New House Internet Services B.V., Rotterdam, NL) to create single high-resolution images for each cross-section.

Measurements of the width of annual growth rings were made on the basis of digital images using WinDENDRO software (Regent Instruments Canada Inc.) at 2–3 radii per section, depending on circularity. Cross-dating was accomplished by applying two levels: first, the different radii within each sample were visually compared, an average of these measurements was created (or a distorted course was rejected if needed), and then dating and measurement errors were checked with the standard dendrochronological software COFECHA. Some of the samples used in the dendrochronological analyses were excluded from further analysis due to the presence of anatomical anomalies, such as wounds, large eccentricity, or inconsistency of the measurement sequence with the population mean. The data obtained are characterised by sufficiently high statistics of the constructed chronology (mean r, rbar, eps). To evaluate multiple detrending methods, we calculated a set of common ring width indices for each individual series using the [dplR] R package[74]. We tested all common detrending and smoothing methods, both data-adaptive and biological approaches, including modified negative exponential curves, generalised negative exponential curves, Hugershoff curves, cubic smoothing spline methods, means, and age-dependent splines. As no or negligible age trends were observed in the juniper ring width series, after different types of standardisation were carried out, the most justifiable was a horizontal straight line and a biological approach using regional curve standardisation (RCS). All the detrended series were averaged using the biweight robust mean to reduce the influence of outliers during the assembly of the standard ring width chronology.

To synchronise the samples, we determined the occurrence of extreme events using anatomical irregularities in growth rings, i.e., blue and frost rings. The so-called 'frost rings' are attributed to collapsed and deformed early or latewood cells and rays, illustrated by irregular cellular areas filled with wound parenchyma resulting from frost during the growing season/boreal summer[54]. Blue rings (BRs) are defined as growth rings with a layer of unlignified axial tracheids visible

in blue after the double-staining thin transverse sections of shrubs rings with Safranin and Astra blue dyes. The cell walls, which are richer in lignin, were stained red, and the cell walls, which are richer in cellulose, were stained blue; thus, BRs reflect incomplete lignification, serving as excellent and sensitive proxies for past rapid cooling[75]. Chronologies of anatomical irregularities were created by assigning these structures to calendar years. In chronologies, all these occurrences were considered yearly, whereas in the text, the position of these structures within the growth ring, size or degree of formation was additionally considered and discussed for selected cases.

#### Meteorological and glaciological data

Climate data from the Narsarsuaq station (WMO code 04270, elevation 30 m a.s.l.) for 1961–2023 (63 years) were used to explore the relationships between climate and growth-ring width records. We analysed available parameters, such as the mean, minimum and maximum air temperatures; atmospheric pressure; total precipitation; highest 24-hour precipitation; number of days of snow cover (>50%) (only 1961–1999); and average cloud cover (only 1961–2008). For comparison, we also utilized a longer series of monthly temperatures over the period of 1784–2013 for Qaqortoq (series based on observations from four sites (Ivittuut, Nanortalik, Narsarsuaq and Paamiut) infilled with regressed values, or, in recent times with values from Qaqortoq Heliport if missing data; sites situated along the southern coast of Greenland) and merged SW Greenland series (based on infilled temperature series from Ilulissat, Nuuk and Qaqortoq situated along the southern and western coasts of Greenland) complied by Cappelen & Vinther[76]. Records from selected stations were among the most comprehensive and reliable available for Greenland[77]. All the data were obtained from the Danish Meteorological Institute (DMI).

The dendrochronological record was also correlated with different seasonal and annual datasets of sea ice. Satellite-derived Arctic sea ice extent data from 1978–2023 were obtained from the National Snow and Ice Data Centre (NSIDC). The presatellite data extending back to 1900 were developed by Walsh et al.[78], Hadley Centre's HadISST dataset, adjusted and recalibrated by Connolly et al.[79]. Notably, there is a long tradition of sea ice observations in our area of interest. The region is characterised by large inherent decadal variability in multiannual ice cover, and analysis back to the 1950s revealed recurrent periods of up to five years with significantly increased ice cover. Ice Patrol Narsarsuaq was established in 1959 on the basis of the abandoned American military airfield in Narsarsuaq in southern Greenland. Ice reconnaissance has been completed 1–3 times a week, covering inshore waters and fjords along southern Greenland.

#### Dendroclimatological analysis and climate reconstruction

To establish the relationship between growth and climate, simple Pearson's correlation and bootstrapped correlation analyses were carried out between the ring width index (RWI) chronology and different climatic elements noted at Narsarsuaq meteorological station for the overlapping period (1961–2013) via the treeclim R package[80]. On the basis of growth–climate response analysis, the primary growth-limiting climatic factor was identified. An 18-month dendroclimatic window starting from May of the previous year to October of the current year, along with seasonal aggregation of the monthly data, was used to identify growth-limiting climatic factors. The moving correlation between the chronology and climate data was also analysed in the treeclim R package to investigate the temporal stability of the dendroclimatic response and to check for possible divergence in the data. Temperature reconstruction in the summer season (June–August, JJA) is theoretically possible; thus, it was targeted for reconstruction. A model for reconstructing the sea ice extent was also tested, but the validation results were weaker than those of the other model. A split-sample validation (with the calibration performed on the last half and validation on the first half) was conducted to check the stability of the

relationships. The full period of 1961–2023 was split into 1961–1992 and 1992–2023 as the calibration and verification periods, respectively. To evaluate the model ability, we used Pearson's correlation coefficient (r), explained variance (R2), reduction in error (RE), the coefficient of efficiency (CE), and Durbin–Watson (DW) statistics. Once the model was judged to be effective and stable, a transfer function was established to reconstruct past June to August temperatures for the period covered by the regional composite tree-ring width chronology, up to the point at which the expressed population signal value of the chronology became less than the threshold value of 0.80. Superposed epoch analysis (SEA) was used to test the significance of a mean shrub growth response to certain events, such as volcanic eruptions. In this study, intervals of 5 years before and after a volcanic eruption were analysed using SEA computation from the dplR package[81,82].

#### Reporting summary

Further information on research design is available in the Nature Portfolio Reporting Summary linked to this article.

### Data availability

Source data are provided as a Source Data file. Additional data related to the current study cannot be publicly available, as this data is part of an ongoing dendrochronological project, but may be made available by the corresponding author (Magdalena Opała-Owczarek; e-mail: magdalena.opala@us.edu.pl) upon request. Other datasets used for this research were derived from the public domain resources. Meteorological data were obtained from the Danish Meteorological Institute (DMI). Satellite-derived Arctic sea ice extent data from 1978–2023 were obtained from the National Snow and Ice Data Centre (NSIDC) and from compilations prepared by Connolly et al.[79]. Ice-core inferred non-sea-salt sulphur records from Greenland is available from Sigl et al.[10]. The palaeoclimate data used for comparisons are sourced from primary publications (Anchukaitis et al.[52]; Andresen et al.[83] Gennaretti et al.[51]; Hantemirov et al.[84]; Kobashi et al.[49]; Kononov et al.[53]; Macias Fauria et al.[85] McKay & Kaufman[50]; Miettinen et al.[48]). Source data are provided with this paper.

### Code availability

Statistical analysis in this study was performed with publicly available packages in R (version 4.4.1).

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

## Acknowledgements

The research was funded by a Polish National Science Centre project no. UMO-2019/35/D/ST10/03137 "Reconstruction of climatic conditions in the Arctic before the period of instrumental measurements on the basis of dendrochronological analysis of tundra dwarf shrubs and historical botanical collections" (M.O-O. and P.O) and Research Excellence Initiative, Priority Research Areas of the University of Silesia in Katowice, Environmental and climate change as a public policy challenge in Europe and worldwide" (M.O-O.). We sincerely thank Olof Ryding and Karen Bach for providing access to the collections and for their support during work in the Herbarium, Natural History Museum, University of Copenhagen, Denmark. We are grateful to the Narsarsuaq Museum for information support. We thank the local communities of Narsarsuaq and Qassiarsuk for their kind support.

## Author contributions

Conceptualization: M.O-O.; Methodology: M.O-O. and U.B.; Investigations: M.O-O. and P.O. with input from Ch.L.; Visualization: P.O. and Ch.L.; Writing–original draft: M.O-O., U.B. and P.O.; Writing–review & editing: M.O-O., U.B. and P.O.; Funding acquisition: M.O-O.

## Competing interests

The authors declare no competing interests.
