## [Transparent Peer Review file · Nature Communications]

500-year paleoclimate record inferred from Greenland Juniper wood contextualizes current climate warming

Corresponding Author: Dr Magdalena Opała-Owczarek

Version 0:

Reviewer comments:

Reviewer #1

(Remarks to the Author)
Dear authors, dear Editor,

thanks for the opportunity to review the manuscript with highly relevant topic. The manuscript is concise yet comprehensive and well written to keep the attention of the reader. Presented results are original and methods used are adequate, although not anyhow innovative. This does not necessarily have to be taken as criticism, methods applied serves the purpose very well and they have been routinely used by community for long time. Impressive is the broad combination of methods (wood anatomy, standard dendroclimatology, inclusion of historical material, etc.) which definitely was very labour demanding. Recent literature which dealt with climate sensitivity of junipers or environmental reconstructions using studied species' annual rings as well as quantitative wood anatomy from the Arctic (including Greenland) is nevertheless almost completely absent. This is very disappointing because a lot has been done on that species in the Arctic and authors should build on that knowledge and discuss it. I also have some concerns regarding introductory, discussion and conclusions part which I list below.

Introductory part to me is concise too much. It provides only the link between climate change and ice melt and contextualize the topic based on three (!) citations only. I think the authors should put more effort here to introduce the complete variety of studied issues (e.g. volcanic eruptions, plant growth...) and link them into meaningful block which in the end proves the need for their research and results.

And here is actually another issue I have with intro part (lines 38 – 41). The paragraph with hypothesis reads quite weak, basically: "develop a chronology and reconstruct climate conditions". If only this is provided in the manuscript (which is not the case!) than it does not merit publication in Nature as it has been done many times before (climate reconstruction) by many other proxy archives. In my opinion, achieved results may offer more with the connection to climate-environmental dynamics and this should be incorporated in hypothesis and in the aim of the study.

Results are well presented but insufficiently discussed. To give the example chapter on extreme years does not include even one literature reference although the authors present findings which should be attempted to interpret ("Interestingly, BRs were more common in the earlier period...").

Conclusions. To me Conclusions should provide upscaled yet informative synthesis of findings (not just their summary though!). I would find it beneficial for the reader to rework this section in this respect or just not include it and foster the result part with more discussion. Plus the first sentence of the conclusions is not truth or bold/disrespectful to the authors who published already a lot on annual rings from polar tundra areas (including Greenland and including juniper shrub). The authors surely provide the longest climate reconstruction based on juniper growth from the Arctic, but similarly or even longer chronologies of juniper were published (with worse Rbar or EPS though) as well as long term dendroenvironmental reconstructions have already been presented. I understand the reason for this first sentence but it needs to be more specific to be fair.

Other concrete comments and concerns:

Line 57-58: I am convinced there are already quite a few juniper studies from the Arctic as well as Greenland in particular which may lack some of the aspects you cover but are closely related to climate or environmental change or even ice melt (GRIS).

Line 68 – 73: dtdo important literature is missing. No objections to discuss the sensitivity of junipers from alpine tundra but many studies have been published on junipers from the Arctic and you do not address those at all in your manuscript as the results are the closest geographically and environmentally.

Line 77: “important ecological significance” – it is vague, please, be concrete.

Line 89 – 92: illogical wording. Better refer to increasing or decreasing sea ice extent consistently and relate it to growth. Now, it swifts in the middle of the sentence.

Line 244 and hereafter: to my understanding juniper is not dwarf shrub (those are shrubs rather from High Arctic like *Salix polaris*). I think better adjective for tundra juniper is prostrate.

Line 287 – 302: you mention very limited research of the species but last work you cite is from 1974. Quite a lot has been published on juniper from SE Greenland since that time. (probably not exactly from the spot you did your field work but still relatively nearby).

Line 322: root collar or the main stems were not sampled from the living individuals, then? Just the branches and than serial sectioning? But from your description it seems that fossil individuals were analysed from the main stem (“were found attached to the ground with remnants of the root system”). Could you explain if this methodologically different approach could not compromise the results?

Line 346: was there any minimal angle between analysed radii?

Line 371: why did you decided to use the meteodata from one station and on the CRU-TS or similar integrated datasets?

Line 549: almost four centuries preservation time of fossil samples since die off is almost unbelievable considering high humidity of the region (600 mm) and relatively high temperatures plus snow directly affecting and melting at the surface. Are you sure about the cross-dating here or could you provide maybe field picture of those three oldest fossil individuals? I may be of course wrong but in such climatic conditions I can imagine fossil junipers 4 centuries old only maybe bellow some “rock roof” at the very steep cliff.

Line 585: Fig. 4 – in figure caption you mention part E and F but those are not in the picture itself.

Line 603: Fig. 5 – in Fig. 1 I counted 13 blue rings but here in Fig. 5 I counted 14 blue rings.

Line 627: sea at Sermilik Fjord is influenced solely by East Greenland Current while your site also by Irminger Current. Would it be possible to use reconstruction from geographically and environmentally closer locality?

Line 631: upper index for 18.

Reviewer #2

(Remarks to the Author)

This study presents a high-resolution dendroclimatological reconstruction of summer temperatures in southern Greenland using shrubs, spanning the period 1526–2023 CE. The research provides valuable insights into Arctic climate variability, particularly during the Little Ice Age (LIA), and assesses the impact of volcanic eruptions using wood anatomical features such as blue rings (BRs) and frost rings (FRs). The findings are significant for understanding recent warming trends in the context of natural variability. However, there are several aspects which require clarification and further discussion. Major revision is needed before the publication.

I have several major concerns regarding this study.

1 stability of the temperature reconstruction. The correlation between temperature and tree-ring width shows significant differences between 1960-1992 ($r=0.34$) and 1992-2023 ($r=0.7$) periods (Figure 4). What causes this instability in the relationship? Supplementary Table 2 also shows that the temperature-width correlation gradually increases with rising temperatures. How might this unstable width-temperature relationship affect the reliability of the 500-year reconstruction?

2 Post-2000 decline trend. Both tree-ring width and reconstructed temperatures show a declining trend after 2000 (Figures 2 and 5). What is the reason for this? While there is evidence of global warming hiatus, what specifically accounts for the temperature decrease in this record? Does this decline appear in observational data? Is there any mismatch between ring width and observed temperatures?

3 Inconsistency in trends. Figure 4A shows increasing ring width after 2000, while Figure 2 shows decreasing width during the same period. What explains this discrepancy?

4 Volcanic signal comparison. Tree rings provide precisely dated annual resolution, and the authors identify extreme cold events coinciding with volcanic eruptions. How do other Greenland records (e.g., ice cores) with potential dating errors capture these volcanic-induced cold periods? I recommend to add superposed epoch analysis (SEA) of volcanic events. Comparison with other Greenland temperature reconstructions. This would better demonstrate the reliability of tree-ring reconstructions.

5. Polar region comparisons. The study would benefit from comparing with tree-ring based temperature reconstructions from other Arctic regions (e.g., Alaska, Russian Arctic) to examine pan-Arctic temperature variability.

Specific comments:

Figure 4 caption doesn't match the figure - the caption references panels A-F but only A-D are shown.

Reviewer #3

(Remarks to the Author)

This is a very interesting study using a fairly groundbreaking set of juniper samples to reconstruct temperature in southern Greenland. Adding paleoclimate chronologies such as this in areas that are typically not covered by the usual paleoclimate proxies is very important for understanding regional variations in climatic trends. I commend the authors on their collection of a both living, dead, and historical samples to create an unprecedented juniper chronology. The research is sound and the manuscript is well written.

One important revision I would request is to include the climate-growth table that is currently in the supplemental information as a figure in the main manuscript. I did not see anywhere else that the list of climatic variables assessed was reported, nor any other indication of the strength of these correlations. I believe this revision is absolutely necessary for the manuscript to stand on its own.

The Laki section seems excessively long. The results are interesting, but could be conveyed much more concisely (probably one paragraph). If the details of this event are a primary focus of the research, it should be included in the introduction and stated as a research goal. Similarly, if assessing the utility of blue rings is a significant goal of the project, then that should be clearly stated in the introduction and the idea of blue rings should be introduced there.

Other more minor revisions I would request:

Page 2, Line 63: "relatively less" than what? Why is this significant?

Page 3, line 70: here is the perfect place to reference the climate-growth table

Page 3, line 77: you state that a positive response to high minimum temperatures has "important ecological significance" but do not elaborate. What is the important significance here?

Page 3, line 93: I am skeptical of the feasibility of reconstructing sea ice with this proxy because sea ice extent does not necessarily have a simple linear relationship with summer temperature.

Page 3, line 99: "the model revealed a 46% variance" does not make sense to me. Do you mean the R^2 was .46, indicating the 46% of the variance was predicted by the model?

Version 1:

Reviewer comments:

Reviewer #1

(Remarks to the Author)

I would like to thank the authors for the large input they had to invest into the revised manuscript. I can clearly state that the long list of my comments and concerns was entirely addressed. I hereby support the publication of the manuscript and I am honoured that I could read this among the first scientists. Good job! Thank you :)

Reviewer #2

(Remarks to the Author)

My main concerns have been addressed. The manuscript is ready for publication. Congratulations.

I have one specific suggestion. In line 98 and Figure 7, Hantemirov et al., 2011 reconstruction was used. In fact, Hantemirov et al., 2022 (Nature Communication, Current Siberian heating is unprecedented during the past seven millennia) reconstruction is longer than Hantemirov et al., 2011, Hantemirov et al., 2022 should be cited and used in Figure 7.

Point-by-point response letter

Dear Editor,

We thank the reviewers for their detailed and constructive comments. Their evaluations greatly contributed to the improvement of our manuscript. All reviewers' comments and suggestions are in black, our responses in blue, and quotations from the revised text in italic.

Magdalena Opała-Owczarek

REVIEWER COMMENTS and AUTHORS' ANSWERS

Reviewer #1 (Remarks to the Author):

Dear authors, dear Editor,

thanks for the opportunity to review the manuscript with highly relevant topic. The manuscript is concise yet comprehensive and well written to keep the attention of the reader. Presented results are original and methods used are adequate, although not anyhow innovative. This does not necessarily have to be taken as criticism, methods applied serves the purpose very well and they have been routinely used by community for long time. Impressive is the broad combination of methods (wood anatomy, standard dendroclimatology, inclusion of historical material, etc.) which definitely was very labour demanding.

We sincerely appreciate the encouraging words and positive feedback on our idea and the efforts. We highly appreciate the reviewer's comments and suggestions for improvements. In the following sections of the review, we have answered all the questions and introduced suggested changes.

Recent literature which dealt with climate sensitivity of junipers or environmental reconstructions using studied species' annual rings as well as quantitative wood anatomy from the Arctic (including Greenland) is nevertheless almost completely absent. This is very disappointing because a lot has been done on that species in the Arctic and authors should build on that knowledge and discuss it. I also have some concerns regarding introductory, discussion and conclusions part which I list below.

We appreciate the reviewer's suggestion to strengthen the literature review. We agree with the reviewer that this section on 'state of art' should be expanded. In the first draft of the manuscript, we included that information, but we have to cut it down for reasons of volume. We thank the reviewer for this comment. We are very pleased to have the opportunity to describe the previous studies in more detail now. We have added the appropriate references to all the contributions on this topic (see our detailed answers below).

Introductory part to me is concise too much. It provides only the link between climate change and ice melt and contextualize the topic based on three (!) citations only. I think the authors should put more effort here to introduce the complete variety of studied issues (e.g. volcanic eruptions, plant growth...) and link them into meaningful block which in the end proves the need for their research and results.

We agree with the reviewer. We have expanded the introduction by adding new sections focusing on other proxies used in paleoclimate research in Greenland: ice cores, lake sediments, and plant growth. We followed the reviewer's suggestion and updated the "Introduction" section with a broader review of prior dendrochronological studies from Greenland, adding key references to

contextualise our work within the existing literature. The need for the development of new dendrochronological data is illustrated in the context of this background:

*“Climate reconstruction from Greenland primarily refers to using ice cores extracted from Greenland's ice sheets to understand past climate conditions (Gkinis et al., 2021). These ice cores contain trapped air bubbles, isotopes, dust, and other compounds that can reveal information about environmental changes over time. Using ice cores has challenges, such as limited geographical coverage (ice sheet), dating uncertainties, and limited temporal resolution (Masson-Delmotte et al., 2006). It should be, however, emphasised that ice cores provide the most comprehensive record of volcanic eruptions that have contributed to periods of cooling (Hammer et al., 1980; Abbott & Davies, 2012; Sigl et al., 2015). The results from the sediment cores recovered from Greenlandic lakes provide an invaluable addition to the findings derived from the ice core samples. The continuous archive of past environmental changes based on these proxy data spans the entire Holocene (Axford et al., 2013; Lasher et al., 2020). Nevertheless, a potential limitation of these studies is the possibility of errors in the interpretation of ^{14}C and ^{210}Pb -dated sediment cores, involving e.g. problems of dating calibration and different sedimentation rates and disturbances (Lusas et al., 2017; Anderson et al., 2019). Tree rings, one of the most widely used climate proxies, are invaluable and powerful sources of climate information due to their high resolution, accurate year-by-year dating, and the ability to reconstruct decadal to millennial-scale climate conditions (LaMarche, 1978). However, the age of the tundra plants has limited the application of the dendrochronological method in Greenland. Existing shrub records of *Alnus viridis*, *Salix glauca* and *Juniperus communis* from West Greenland (Jørgensen et al., 2015; Hollesen et al., 2015; Lehejček et al., 2017; Wilmking et al., 2018; Prendin et al., 2022), *Salix arctica* from north-eastern Greenland (Schmidt et al., 2006) or from the only natural forest with *Betula pubescens* located in Qinngua Valley, southern Greenland (Kuivinen & Lawson, 1982; Xu et al., 2021) do not reach the pre-instrumental period. Until now, the existing dendrochronological data for Greenland have been insufficient for reconstructing the climate over the centuries.”*

And here is actually another issue I have with intro part (lines 38 – 41). The paragraph with hypothesis reads quite weak, basically: “develop a chronology and reconstruct climate conditions”. If only this is provided in the manuscript (which is not the case!) than it does not merit publication in Nature as it has been done many times before (climate reconstruction) by many other proxy archives. In my opinion, achieved results may offer more with the connection to climate-environmental dynamics and this should be incorporated in hypothesis and in the aim of the study.

The reviewer is right that this issue should be addressed more thoroughly. In the revised manuscript, we extended the "Introduction" section: “Here, we present the combined dendrochronological and wood anatomical assessment of living and relict juniper wood from southern Greenland to reveal annually resolved and absolutely dated insights about natural summer temperature variability back into the Little Ice Age. This reconstruction extends beyond the instrumental period in Greenlandic terrestrial areas outside the ice sheet. Additionally, we aimed to assess the usefulness of the so-called “blue rings” as an indicator of post-volcanic cooling and as a key anatomical feature allowing for the synchronisation of juniper samples. Our study contributes to the understanding of the effects of volcanic eruptions on climate by combining wood anatomical observation with superposed epoch analysis to explore the significance of selected eruptions on growth ring width and blue ring formation. In order to more accurately evaluate the reliability of the reconstruction, we conducted thorough comparisons with other proxy data from southern Greenland and dendroclimatic reconstructions from the Arctic.”

Results are well presented but insufficiently discussed. To give the example chapter on extreme years does not include even one literature reference although the authors present findings which should be attempted to interpret (“Interestingly, BRs were more common in the earlier period...”).

We agree with the reviewer. For clarity, we removed the first sentence of this chapter, and instead we added a comparison with literature, as suggested:

“Extreme years are manifested by the occurrence of narrow rings or disrupted cell formation, such as BRs and FRs (Fig. 5). Most of these extreme years, associated with volcanic eruptions, are consistent with studies carried out at the northern tree limit in both Asia and North America, e.g. 1600, 1641, 1783, 1815-1817, 1868, 1884, 1912, 1916-1917 (LaMarche & Hirschboeck, 1984; Filion et al., 1986; Delwaide et al., 1991; Briffa et al., 1998; Hantemirov et al., 2004).”

According to the reviewer's suggestion, we extended the discussion: “In general, BRs appear to occur during warm periods, such as the 1720–1820 period and the modern warm period. In contrast, FRs occur more frequently during the cool episodes of the Little Ice Age. These results corroborate the hypothesis that prolonged cooling does not diminish lignification, but rather increases the likelihood of frost-related damage. These observations coincide with the results of studies conducted at the upper tree line in the Swiss Alps (Körner et al., 2023).”

Conclusions. To me Conclusions should provide upscaled yet informative synthesis of findings (not just their summary though!). I would find it beneficial for the reader to rework this section in this respect or just not include it and foster the result part with more discussion.

We appreciate the reviewer's suggestion to rework the conclusions. In response, we added new information that synthesises our findings and provide a broader context for our achievements:

“... The unique results obtained through the use of specimens from historical scientific expeditions, currently preserved in herbaria, highlight the importance of reanalysing old botanical collections using modern analytical techniques. Since the area of our research, located within Eriksfjord, was inhabited by the Vikings from 985 CE to the mid-15th century (Eastern Settlement), research into past climate change is becoming increasingly important. Integrating wooden archaeological finds from this area with the established five-century-long juniper growth ring chronology will be the next step in reconstructing past changes in this area over a period of environmental changes at the turn of the 15th and 16th centuries.”

“...The analysis of these anatomical changes has allowed for a significant deepening of conclusions.”
“Despite numerous tephra deposits, meteorological data and historical accounts indicating a harsh summer in 1783 and a severe winter 1783/1784, the blue rings are the first biological evidence of climate disturbances associated with the Laki volcano eruption in the Arctic. Detecting these anatomical features will provide a more complete and accurate understanding of the eruption's dynamics and atmospheric effects, and will help to create a useful model for assessing the impact of past volcanic activity on the Earth's climate.”

Plus the first sentence of the conclusions is not truth or bold/disrespectful to the authors who published already a lot on annual rings from polar tundra areas (including Greenland and including juniper shrub). The authors surely provide the longest climate reconstruction based on juniper growth from the Arctic, but similarly or even longer chronologies of juniper were published (with worse Rbar or EPS though) as well as long term dendroenvironmental reconstructions have already been presented. I understand the reason for this first sentence but it needs to be more specific to be fair.

We agree with the reviewer, we should have been clearer about this in the original manuscript. We modified this sentence by adding “...which have not been *entirely* represented in long-term dendroclimatological reconstructions”, and “The developed record is ~~the~~ *one of the longest growth ring chronologies for Arctic shrubs to date*”. We have added information about other works on juniper, including works from Greenland, to the body of the article and to the references list.

Other concrete comments and concerns:

Line 57-58: I am convinced there are already quite a few juniper studies from the Arctic as well as Greenland in particular which may lack some of the aspects you cover but are closely related to climate or environmental change or even ice melt (GRIS).

We changed this part according to the reviewer's suggestion and added relevant citations. Some of the most recent work was not known at the time the manuscript was prepared, but we can now add it. We agree with the reviewer that this section should be expanded. Thanks to the editor's decision, we can now expand this paragraph accordingly. Now is: “*In only a few dendrochronological studies to date, it has been possible to assemble a multi-century chronology of junipers from Arctic sites, of which the longest, based on both living and dry branches, comes from the Kola Peninsula (>500 yrs, Shumilov et al. 2007), Polar Urals (>600 yrs, Hantemirov et al. 2011), and northern Iceland (>800 yrs, Opała-Owczarek et al. 2025). Recently, very old specimens of junipers have also been described for sites in northern Fennoscandia (Sør-Varanger, Kevo, Abisko) by Lehejček et al. (2024) and Carrer et al. (2025). The assembled juniper chronologies for Greenland mainly concerned the area around Nuuk, where the relation to climate and Greenland Ice Sheet melting was elaborated over the last century (Buras et al. 2017 and Lehejček et al. 2017). Juniper discs with more than 200-300 rings were collected from southernmost Greenland during the scientific expeditions to Greenland at the end of the 19th century (Meddelelser om Grønland..., 1888), and have recently been reconfirmed.*”

Line 68 – 73: The important literature is missing. No objections to discuss the sensitivity of junipers from alpine tundra but many studies have been published on junipers from the Arctic and you do not address those at all in your manuscript as the results are the closest geographically and environmentally.

We agree with the reviewer that this section should be expanded. Thanks to the editor's decision we can now expand our text and add this relevant information: “*Similarly, junipers from other circum-Arctic sites are temperature-stressed (Hallinger et al., 2010; Hantemirov et al. 2011; Lehejček et al. 2017; Opała-Owczarek et al. 2025), however, the strength of the signal varies depending on inter alia sampling strategy (adequate replication, proximity to the root collar) and site characteristics (homogeneity, geomorphic disturbances).*”

Line 244 and hereafter: to my understanding juniper is not dwarf shrub (those are shrubs rather from High Arctic like *Salix polaris*). I think better adjective for tundra juniper is prostrate.

Thank you for this comment. We referred to “dwarf juniper”, not „dwarf shrub”, but for clarity we deleted “dwarf”.

Line 287 – 302: you mention very limited research of the species but last work you cite is from 1974. Quite a lot has been published on juniper from SE Greenland since that time. (probably not exactly from the spot you did your field work but still relatively nearby).

Thank you for this comment. We agree with the reviewer that this section should be expanded. Thanks to the editor's decision, we can now expand our text and add these relevant citations:

“*.... which was also confirmed later by Buras et al. (2017) and Lehejček et al. (2017), who examined the potential of shrub ring width and wood anatomy as proxies for climate and GrIS-melt. Tumajer et al. (2021) point to the intensification of growth rates during the short growing seasons in the Arctic.*”

Line 322: root collar or the main stems were not sampled from the living individuals, then? Just the branches and then serial sectioning? But from your description it seems that fossil individuals were analysed from the main stem (“were found attached to the ground with remnants of the root system”). Could you explain if this methodologically different approach could not compromise the results?

We apologise for the unclear description. We did not take samples from the root collars because the growth disturbances present there may negatively influence the dendroclimatic signal. Our experience suggests that samples from the root collar are important when studying polar willows, dwarf birch and other dwarf shrubs. Our aim was not to check the maximum age of the plant, but to make the most accurate synchronisation between samples and a robust climate reconstruction. We should have been clearer about this in the original manuscript. We took samples from the main stems. For clarity, we modified this description; now it is: “Straight sections of the main prostrate stems were sampled for analysis to exclude the possible presence of reaction wood. Samples were also taken from contemporary growing specimens for calibration with meteorological data. For these specimens, the discs from one of the main stems were collected to avoid destroying the whole plant.”

Line 346: was there any minimal angle between analysed radii?

No minimum angle was set, but measurement paths were arbitrarily chosen to best represent the growth over the entire cross-section. This was dependent on the appearance of each cross-section.

Line 371: why did you decide to use the meteorological data from one station and on the CRU-TS or similar integrated datasets?

In the case of Greenland, a single grid can cover very different environments, covering part of the ice sheet, part of the sea area and part of the land area. Our aim was to reconstruct the climate of the unglaciated land area (based on terrestrial proxy) as the only one of its kind for Greenland. In view of this, in order to represent the relationship between shrub ring width and climate conditions as accurately as possible, we took into account complete, homogeneous data for different climate elements, from the meteorological station Narsarsuaq closest to our sampling sites. We made a number of comparisons, e.g. between the two official nearest met stations, Narsarsuaq and Qaqortoq, located within the fjord and near open sea, which shows variation in the distribution of meteorological elements throughout the year. Station data are much more precise, but they could represent very unique (particular) local conditions. In the future, spatial analyses using dendrochronological data from the whole of coastal Greenland should be considered. The overarching aim of our work is not spatial analyses, which, due to their volume and multi-faceted nature, require a separate article.

Line 549: almost four centuries preservation time of fossil samples since die off is almost unbelievable considering high humidity of the region (600 mm) and relatively high temperatures plus snow directly affecting and melting at the surface. Are you sure about the cross-dating here or could you provide maybe field picture of those three oldest fossil individuals? I may be of course wrong but in such climatic conditions I can imagine fossil junipers 4 centuries old only maybe below some “rock roof” at the very steep cliff.

We realise that these results are surprising. However, note that we only included three such old samples in the chronology, as these were characterised by very high statistical metrics (cofecha output “correlation of series by segments” between 1525 – 1649 was: .58, .52, .58, .59); thus, we are confident of the dating result. We also enclose below the pictures of the oldest sample, another example is visible on figure 1B. It is indeed not known whether all the rings on these samples are

present due to their state of preservation, but the preserved ring sequences were long enough to be dated.

The possibility of finding such old specimens of juniper branches in the field is also confirmed by Russian (Hantemirov et al., 2004) and Icelandic (Opała-Owczarek et al., 2025) works. Recently, similar dating of dead juniper branches was shown by Carrer et al. (2025) for samples from Kevo, Finland (last year on sample 1469) and Chernaya, Russia (last year on sample 1438).

Archaeological finds from Viking-era sites in Greenland, Iceland and the Faroe Islands also contain juniper wood. The existence of archaeological objects and samples from juniper wood also proves that it can survive in the field for several hundred years (both in a bad and good state of preservation). This is confirmed by our observations of archaeological wood from Greenland (unpublished data). For example, counting sticks, part of a decoration, as well as charcoals, were deposited ca. 1000 AD. Other samples preserved in the waterlogged layers were in a much better state of preservation due to the constantly cold, wet and anaerobic conditions (dated AD 770-1015). Wood of conifers and dwarf shrubs was much better preserved than the wood of larger broad-leaved trees, which was generally more decayed. The juniper wood, used for ropes, bucket loops, and short pointed sticks, is common on Viking Age sites. Descriptions of such artefacts and their archaeological dating have been provided by, among others, Larsen (1991), Malmros (1994), Pinta (2018), Guðmundsdóttir (2021), and Mooney et al. (2022).

Data from the literature indicate that juniper is a very resistant species, as discussed by Opała-Owczarek et al. (2025): In the context of establishing the composite juniper chronology, it is crucial to recognise the unique preservation of relict juniper wood in the cold environments. Juniper growing in such conditions exhibits a notably high natural resistance to decay (Scheffer & Morrell, 1998; Morrell et al., 1999; Opała-Owczarek et al., 2018). Its wood is characterised by high density and durability, which is related to harsh ecological conditions. The juniper wood is also extremely durable in contact with the soil, little shrinks and does not tend to crack. Furthermore, it displays excellent resistance to fungal attack (*Xylophagous* fungi) (Morrell et al., 1999). Even after contact with water does not decay, and after drying do not reduce their size (Mukhamedshin and Talancev, 1982). In semi-arid cold ecosystems, the time of wood decay is significantly longer than in wet areas (Murphy et al., 1998).

In addition to strictly scientific research, practical studies have been carried out on preserving dry juniper, which indicates its exceptional resistance to climatic conditions. Experimental pole farm

tests conducted in the Cascade Mountains in Oregon revealed that juniper exhibits exceptional natural durability (Morrell et al., 1999). It should be noted that the test studies were carried out in an area with a mild and humid climate (average annual precipitation 1050 mm), typically experiencing dry summers and rainy winters. This climate favours the growth of wood-destroying organisms throughout the year.

Guðmundsdóttir (2021), Wood procurement in Norse Greenland (11th to 15th c. AD), *Journal of Archaeological Science* 134 (2021) 105469

Larsen (1991), Norsemen's use of juniper in Viking Age Faroe Islands. *Acta Archaeologica*, 61, 60-72.

Malmros, C., 1994. Exploitation of local, drifted and imported wood by the vikings on the Faroe Islands. *Bot. J. Scotl.* 46, 552–558.

Mooney et al. (2022), Wood resource exploitation in the Norse North Atlantic: a review of recent research and future directions, *Expanding Horizons UBAS* 13.

Morrell, J.J., Miller, D.J., Schneider, P.F., 1999. Service life of treated and untreated fence posts: 1996 post-farm report. Forest Research Laboratory. Oregon State University. Research Contribution 26. 24 pp.

Pinta (2018), Norse Management of Wooden Resources across the North Atlantic: Highlights from the Norse Greenlandic Settlements, *Environmental Archaeology*, DOI: 10.1080/14614103.2018.1547510

Line 585: Fig. 4 – in figure caption you mention part E and F but those are not in the picture itself.

We apologise for this oversight. Captions A and B referred to the results, which appeared in the supplementary materials (as Table 2). Now it was all properly compiled in Figure 4, with appropriate captions from A to F.

Line 603: Fig. 5 – in Fig. 1 I counted 13 blue rings but here in Fig. 5 I counted 14 blue rings.

Thank you for this comment. We apologise for this oversight. It should be 13 BR on both graphs. We corrected the mistake on Fig.5, where in 1816 should be FR, not BR.

Line 627: sea at Sermilik Fjord is influenced solely by East Greenland Current while your site also by Irminger Current. Would it be possible to use reconstruction from geographically and environmentally closer locality?

We made comparisons with the existing reconstructions for the Sermilik fjord, as they had a relatively higher resolution and thus could be compared with our dendrochronological reconstruction. In the vicinity of our site there are only reconstructions inferred from Lake Igaliku sediments. In response to this comment, we added chironomid assemblages stratigraphy, as a proxy of climate variability, to Fig. 6. We also added sentences to the text: "In the closest vicinity of the juniper study sites, climate variability was previously inferred from Lake Igaliku sediments (Millet et al., 2014). A comparison of the data obtained from the chironomid assemblages with the juniper-based reconstruction reveals a number of similarities, except for the periods between 1640 and 1790, and between 1920 and 1970. The most consistent signals are the strong modern warming and the relatively warm conditions during the Dalton Minimum and the warming of the early 17th century (Fig. 6E)."

Line 631: upper index for 18.

Thank you. Corrected.

Reviewer #2 (Remarks to the Author):

This study presents a high-resolution dendroclimatological reconstruction of summer temperatures in southern Greenland using shrubs, spanning the period 1526–2023 CE. The research provides valuable insights into Arctic climate variability, particularly during the Little Ice Age (LIA), and assesses the impact of volcanic eruptions using wood anatomical features such as blue rings (BRs) and frost rings (FRs). The findings are significant for understanding recent warming trends in the context of natural variability. However, there are several aspects which require clarification and further discussion. Major revision is needed before the publication.

Thank you for appreciating our idea and our efforts. We are very grateful for your suggestions for additional analyses, which will certainly enrich this text. In the following sections of the review, we have addressed all the questions and comments, as well as incorporated the suggested additions.

I have several major concerns regarding this study.

1 stability of the temperature reconstruction. The correlation between temperature and tree-ring width shows significant differences between 1960-1992 ($r=0.34$) and 1992-2023 ($r=0.7$) periods (Figure 4). What causes this instability in the relationship? Supplementary Table 2 also shows that the temperature-width correlation gradually increases with rising temperatures. How might this unstable width-temperature relationship affect the reliability of the 500-year reconstruction?

We thank the reviewer for this insightful comment and for paying attention to the stability of the temperature-width correlation. We agree that this issue is particularly important, as reduced sensitivity of tree growth to temperature has been evidenced at high northern latitudes and alpine locations. In our research, we do not observe an issue of "Divergence Problem". And the designated dendroclimatic model indicates a strong temperature signal, especially in recent decades.

We performed the JJA temperature reconstruction based on data from the Narsarsuaq station, but in developing the reconstruction model, we also tested for data from Qaqortoq. Thank you for pointing out the inconsistency in the figures. By an editorial mistake, the statistics for the model (measures of reconstruction skill: r , RE, CE, and ST) appeared with the data from Qaqortoq. We have now corrected this, and the respective data for Narsarsuaq are presented. For clarity we also added: "...noted at Narsarsuaq meteorological station" to the Dendroclimatological analysis and climate reconstruction section.

Once again, we have re-checked the source data. Indeed, differences between temperature and tree-ring width are observed, but this is evident mostly for Qaqortoq station. These lower correlation values in the earlier period can be observed in Fig. 4B. The Qaqortoq series, as we indicated in the Material paragraph, is based on observations from four sites (Ivittuut, Nanortalik, Narsarsuaq and Paamiut), infilled with regressed values, or, in recent times, with values from Qaqortoq Heliport. Additionally, Qaqortoq station was relocated in September of 2003, and earlier location/relocations are not known or certain (Cappelen, 2014; Cappelen and Vinther, 2014). These elements may have influenced the quality of the data in the earlier period. For Narsarsuaq, these correlations are more stable over time, which has to do with the better quality and continuity of instrumental data from the Narsarsuaq airport.

Although the data for Narsarsuaq are homogeneous, slightly lower r value in the earlier period is apparent. Other researchers have pointed to the possibility of insect gradation affecting the stability of the climate signal from south-eastern Greenland (Wilmking et al., 2018), but the juniper species that we studied was not infected. Perhaps another environmental factor contributed to minor inaccuracies in the chronology and instrumental data (e.g. 1973–1982), but we are unable to identify them unequivocally. However, this is a short period of time, and in our opinion, this does not

significantly affect the overall reconstruction, and the model is characterised by good measures of reconstruction skills. As Cappelen (2014) points out, southern Greenland is characterised by a very large variation in weather conditions, which can affect the stability of the climate signal. This is due to the very strong interplay between sea and land areas in close proximity to the ice sheet. The large temperature differences in the area – between the cold sea and the warm inland area in the summer and between the warm sea and the cold inland area in the winter – give rise to a local but dominant monsoon system in the fjords. This pattern is often disrupted during periods of unstable weather.

2 Post-2000 decline trend. Both tree-ring width and reconstructed temperatures show a declining trend after 2000 (Figures 2 and 5). What is the reason for this? While there is evidence of global warming hiatus, what specifically accounts for the temperature decrease in this record? Does this decline appear in observational data? Is there any mismatch between ring width and observed temperatures?

We appreciate the reviewer's suggestion to strengthen the discussion on this valid issue, so we added supplementary information to the text: *"The meteorological and tree ring data are consistent and show that the southern part of Greenland experienced decadal periods of both cooling and warming during 1961–2023, with an inflection point around the mid-1990s, and no significant warming after ~2010 (Suppl. Fig. 1). The reasons for this variability can be found in the regional influence of the large-scale circulation, represented by the indices for the NAO (North Atlantic Oscillation) and GBI (Greenland blocking index). Zhang et al. (2022) concluded that, since 2011, there has been a shift towards a more negative phase for GBI and a more positive phase for NAO, but at the same time, these have been highly variable, with more modest values and reduced warming recently."*

We also added new reference:

Zhang, Q., Huai, B. J., Van Den Broeke, M. R., Cappelen, J., Ding, M. H., Wang, Y. T., & Sun, W. J., 2022: Temporal and spatial variability in contemporary Greenland warming (1958–2020). *Journal of Climate*, 1(9), 2755–2767. <https://doi.org/10.1175/JCLI-D-21-0313.1>

3 Inconsistency in trends. Figure 4A shows increasing ring width after 2000, while Figure 2 shows decreasing width during the same period. What explains this discrepancy?

The data series shown in these graphs is the same. The visual impression that the trend is different is due to the fact that the data in Fig.2. covers the entire period of the five centuries. While the data series in Fig. 4a covers the last 100 years, and is truncated to 2020 to be consistent with the ice extent data. The last 5 years have been a period of decreasing juniper growth-ring width.

4 Volcanic signal comparison. Tree rings provide precisely dated annual resolution, and the authors identify extreme cold events coinciding with volcanic eruptions. How do other Greenland records (e.g., ice cores) with potential dating errors capture these volcanic-induced cold periods? I recommend to add superposed epoch analysis (SEA) of volcanic events. Comparison with other Greenland temperature reconstructions. This would better demonstrate the reliability of tree-ring reconstructions.

This is an excellent comment, although in the primary version of the text, we did not plan to include SEA analysis. We followed the reviewer's suggestion and compared the BR signal in the Greenland juniper with the volcanism recorded in the Greenlandic ice cores. We have added to Figure 5 a panel showing the amount of sulphur from NEEM ice core. Taking into account the reviewer's suggestion, we also made an additional comparison between the occurrence of dendrochronological signals of eruptions (visible as blue rings 'BR'), with the increased amount of non-sea-salt sulphur records from Greenland for the NEEM ice core (after Sigl et al. 2015). According to the suggestion, we also

performed superposed epoch analysis (SEA) of volcanic events, which is now presented as lower panel in Fig. 5.

Beyond the new elements added to Figure 5, we added relevant comments in the text: *“We evaluated the accuracy of our dendrochronological data by comparison with volcanic signals from the Greenland ice sheet (Sigl et al. 2015). Volcanic aerosol peaks contained in the ice core (NEEM-2011-S1) are consistent with dendrochronological signals (narrow rings, BRs, FRs), which are closely related to major volcanic eruptions (Fig.5 A, B). This is particularly pronounced in years of volcanic eruptions: Huaynaputina (1600), Mt. Parker (1641), Hekla (1963), Laki (1783), Tambora (1815), and Mt.Katmai (1912). These events were followed by the occurrence of BRs, FRs or strong growth reduction in tree-ring records. Superposed epoch analyses show a clear post-volcanic cooling signal after most major eruptions, or a response lagged by one year (Fig. 5C). Of particular interest is the response of Greenland's junipers to the major eruptions of Laki and Tambora, where BRs were recorded in the year of the eruption and narrow growth a year later (Fig. 5Ca).”*

In Figure 6, we additionally show a comparison with different temperature reconstructions from Greenland. We have also added new sentences to the paragraph with multi-proxy comparison: *“In the closest vicinity of the juniper study sites, climate variability was previously inferred from Lake Igaliku sediments (Millet et al. 2014). A comparison of the data obtained from the chironomid assemblages with the juniper-based reconstruction reveals a number of similarities, except for the periods between 1640 and 1790, and between 1920 and 1970. The most consistent signals are the strong modern warming and the relatively warm conditions during the Dalton Minimum and the warming of the early 17th century.”*

5. Polar region comparisons. The study would benefit from comparing with tree-ring based temperature reconstructions from other Arctic regions (e.g., Alaska, Russian Arctic) to examine pan-Arctic temperature variability.

Thank you for these suggestions. According to the reviewer's advice, we added comparisons with the existing reconstructions from northern Quebec, northern Alaska, the Polar Urals and the Kola Peninsula to the newly created Figure 7. We also added a few sentences with discussion: *“Comparisons of the new dendrochronological record from southern Greenland with dendrochronological data from the northern tree lines of Eurasia and North America revealed similarities and significant differences in pan-Arctic temperature variability (Fig. 7). The greatest agreement was found in the Northern Quebec temperature reconstruction (Gennaretti et al., 2017), which shows similar fluctuations in both warm (recent warming; turn of the 18th and 19th centuries; first half of the 18th century) and cool (1840s–1950s) periods. By contrast, the common warm periods in Greenland and Alaska (Anchukaitis et al., 2013) are the recent warming period, the 18th century, and the mid-16th century. The most notable inconsistency concerns the Maunder Minimum period, which is distinctly cold in many reconstructions, but not clearly marked in Greenland. Similarly, this MM cooling is absent in data from the Russian Arctic (Hantemirov et al., 2011). The reconstruction of temperature variability from the Kola Peninsula (Kononov et al., 2009) differs most significantly, with only the warming and subsequent cooling in the 17th century being consistent with that observed in southern Greenland (Fig. 7).”* We also added relevant citations to the Reference list:

Anchukaitis, K.J., D'Arrigo, R.D., Andreu-Hayles, L., Frank, D., Verstege, A., Curtis, A., Buckley, B. M., Jacoby, G. C., Cook, E. R. (2013). Tree-ring-reconstructed summer temperatures from Northwestern North America during the last nine centuries, *J. Clim.*, 26 (10), 3001–3012.

Gennaretti, F., Huard, D., Naulier, M. et al. (2017). Bayesian multiproxy temperature reconstruction with black spruce ring widths and stable isotopes from the northern Quebec taiga. *Clim Dyn* 49, 4107–4119.

Hantemirov R., Shiyatov S., Gorlanova L. (2011). Dendroclimatic study of Siberian juniper, *Dendrochronologia* 29, 119–122.

Kononov, Y. M., Friedrich, M., Boettger, T., (2009). Regional Summer Temperature Reconstruction in the Khibiny Low Mountains (Kola Peninsula, NW Russia) by Means of Tree-ring Width during the Last Four Centuries, *Arctic, Antarctic, and Alpine Research*, 41:4, 460-468.

Specific comments:

Figure 4 caption doesn't match the figure - the caption references panels A-F but only A-D are shown. **Thank you for this remark. We apologise for this oversight. Captions A and B referred to the results, which appeared in the supplementary materials (as Table 2). Now it was all properly compiled in Fig.4, with appropriate captions from A to F.**

Reviewer #3 (Remarks to the Author):

This is a very interesting study using a fairly groundbreaking set of juniper samples to reconstruct temperature in southern Greenland. Adding paleoclimate chronologies such as this in areas that are typically not covered by the usual paleoclimate proxies is very important for understanding regional variations in climatic trends. I commend the authors on their collection of a both living, dead, and historical samples to create an unprecedented juniper chronology. The research is sound and the manuscript is well written.

We sincerely thank the reviewer for the positive feedback and encouraging words.

One important revision I would request is to include the climate-growth table that is currently in the supplemental information as a figure in the main manuscript. I did not see anywhere else that the list of climatic variables assessed was reported, nor any other indication of the strength of these correlations. I believe this revision is absolutely necessary for the manuscript to stand on its own.

We followed the reviewer's suggestion and have added the climate-growth table to our main text, as part of Figure 4.

The Laki section seems excessively long. The results are interesting, but could be conveyed much more concisely (probably one paragraph). If the details of this event are a primary focus of the research, it should be included in the introduction and stated as a research goal. Similarly, if assessing the utility of blue rings is a significant goal of the project, then that should be clearly stated in the introduction and the idea of blue rings should be introduced there.

The reviewer is correct that this part can be more concise. We have shortened this paragraph (by around 90 words) as suggested. We have also added information about the utility of blue rings, and the Laki case study is also added as a significant goal to the introduction: *“Additionally, we aimed to assess the usefulness of the so-called “blue rings” as an indicator of post-volcanic cooling and as a key anatomical feature allowing for the synchronisation of juniper samples. Our study contributes to the understanding of the effects of volcanic eruptions on climate by combining wood anatomical observation with superposed epoch analysis to explore the significance of selected eruptions on growth ring width and blue ring formation.”*

Other more minor revisions I would request:

Page 2, Line 63: "relatively less" than what? Why is this significant?

The reviewer is correct that this should be clarified. We changed to "significantly less". The lower number of blue rings detected in Greenlandic juniper than in other studies is favourable for dating, because if there are too many, it is difficult to find a common pattern in their occurrence.

Page 3, line 70: here is the perfect place to reference the climate-growth table

We have added the reference to climate-growth results (now incorporated in Figure 4, as A,B panels).

Page 3, line 77: you state that a positive response to high minimum temperatures has "important ecological significance" but do not elaborate. What is the important significance here?

For clarity, we removed this sentence, and instead we have added a more concise description, and a reference to earlier work on deciduous shrubs from Greenland, as requested by other reviewer: "The positive correlation between winter-spring minimum temperature and juniper growth corresponds with the negative correlation with snow cover, indicating that less snow cover and faster snowmelt have a positive effect on juniper growth. However, it should be noted that precipitation was not statistically significant, as plants have access to moisture through both precipitation and high air humidity. So, long-lasting snow cover is unfavourable to juniper growth, thereby significantly shortening the length of the growing period during a short Arctic summer. These results concur with findings from other parts of Greenland where deciduous shrubs exhibit a consistent negative response towards the amount of snow precipitation (Schmidt et al. 2010; Gamm et al. 2018)."

Page 3, line 93: I am skeptical of the feasibility of reconstructing sea ice with this proxy because sea ice extent does not necessarily have a simple linear relationship with summer temperature.

Thank you for this comment. We supplemented the sentence: "However, attempts to reconstruct this element have been unsuccessful and require future study, as sea ice extent does not necessarily have a simple linear relationship with summer temperature."

Page 3, line 99: "the model revealed a 46% variance" does not make sense to me. Do you mean the R^2 was .46, indicating the 46% of the variance was predicted by the model?

Thank you for this comment. For clarity, we modified it in the text: "The coefficient of determination was 0.46, indicating that 46% of the variance was predicted by the model..."

Point-by-point response letter 2

Dear Editor,

We sincerely appreciate recognition of our work. We have implemented a minor suggestion from one of the reviewers. We have also responded to all of your technical comments.

Magdalena Opała-Owczarek

REVIEWERS' COMMENTS and AUTHORS' ANSWERS

Reviewer #1 (Remarks to the Author):

I would like to thank the authors for the large input they had to invest into the revised manuscript. I can clearly state that the long list of my comments and concerns was entirely addressed. I hereby support the publication of the manuscript and I am honoured that I could read this among the first scientists. Good job! Thank you :)

Thank you very much for appreciating our work.

Reviewer #2 (Remarks to the Author):

My main concerns have been addressed. The manuscript is ready for publication. Congratulations. I have one specific suggestion. In line 98 and Figure 7, Hantemirov et al., 2011 reconstruction was used. In fact, Hantemirov et al., 2022 (Nature Communication, Current Siberian heating is unprecedented during the past seven millennia) reconstruction is longer than Hantemirov et al., 2011, Hantemirov et al., 2022 should be cited and used in Figure 7.

We sincerely appreciate your recognition of our work. We have taken into account the proposed change in citation. Thank you.